# Genetic Effects of Single Nucleotide Polymorphisms in the Goat *GDF9* Gene on Prolificacy: True or False Positive?

**DOI:** 10.3390/ani9110886

**Published:** 2019-10-31

**Authors:** Xinyu Wang, Qing Yang, Sihuan Zhang, Xiaoyu Zhang, Chuanying Pan, Hong Chen, Haijing Zhu, Xianyong Lan

**Affiliations:** 1College of Animal Science and Technology, Northwest A&F University, Shaanxi Key Laboratory of Molecular Biology for Agriculture, Yangling 712100, China; wangxinyu6157@163.com (X.W.); yangqing199506@163.com (Q.Y.); sihuanzhang1990@163.com (S.Z.); zhangxiaoyu728@163.com (X.Z.); chuanyingpan@126.com (C.P.); chenhong1212@126.com (H.C.); 2Shaanxi Provincial Engineering and Technology Research Center of Cashmere Goats, Yulin University, Yulin 719000, China; 3Life Science Research Center, Yulin University, Yulin 719000, China; 4School of Medicine, Sun Yat-sen University, Guangzhou 510080, China; 5School of Life Sciences and Biotechnology, Shanghai Jiao Tong University, Shanghai 200030, China

**Keywords:** goat, *GDF9* gene, SNP, reproductive traits, association

## Abstract

**Simple Summary:**

As an important regulator factor, which was secreted by female oocytes, the growth differentiation factor 9 (*GDF9*) plays an essential role during the growth and differentiation of ovarian follicles. Single nucleotide polymorphisms (SNPs) within the *GDF9* gene have been found to be involved in reproductive traits in livestock, and some of these mutations have been used as the effective makers in animal molecular breeding. However, it is remarkable that the SNPs of the goat *GDF9* gene have not been systematically sorted and analyzed from the reported studies, which leads to an inability to find effective loci that could be applied in improving the prolificacy of goats via the molecular breeding method. In this study, we gathered and sorted 45 SNPs of the goat *GDF9* gene from all relevant studies and the National Center for Biotechnology Information Search database (NCBI), and especially analyzed and discussed the relationship between part controversial and potentially effective SNPs and the reproductive traits. The results indicated that non-synonymous SNPs A240V, Q320P, and V397I and synonymous SNPs L61L, N121N, and L141L were six “true” positive SNPs in improving goat fertility. Nevertheless, the regulation pathways and the specific mechanism of these six SNPs on goat fecundity are not clear, which still need further study in more goat breeds and a large sample size. These results provided an effective tool for follow-up research studies on the molecular genetic breeding of goats’ reproductive traits.

**Abstract:**

Goat reproductive traits are complex quantitative traits controlled by polygenes and multipoint. To date, some high-fertility candidate genes in livestock have been unearthed and the growth differentiation factor 9 (*GDF9*) gene is one of them, which plays a crucial role in early folliculogenesis. According to the relevant previous studies and the National Center for Biotechnology Information Search database (NCBI), a total of 45 single nucleotide polymorphisms (SNPs) have been detected in the goat *GDF9* gene, but which one or which ones have important effects on goat fecundity is still uncertain. Hence, in order to find effective molecular markers for goat genetic breeding and accelerate the goat improvement, this study summarized and classified the above 45 SNPs into four kinds, as well as compared and analyzed the same SNP effects and the different SNPs linkage effects on the reproductive traits in different goat breeds. Since there were many SNPs in the goat *GDF9* gene, only 15 SNPs have been identified in more than 30 goat breeds worldwide and they showed different effects on the litter size. Therefore, this study mainly chose these 15 SNPs and discussed their relationship with goat productivity. Results showed that three non-synonymous SNPs A240V, Q320P, and V397I and three synonymous ones L61L, N121N, and L141L played a “true” role in the litter size trait in many goat breeds around the world. However, the regulatory mechanisms still need further research. These results provide an effective tool for follow-up research developing the goat molecular breeding strategies and improving the goat reproductive traits.

## 1. Introduction

As one of the earliest domesticated animals, goats have an excellent production performance and their products are ubiquitous and popular all over the world. Goat milk has the characteristics of richer casein micelles, less lactose, and higher proportion of small milk fat globules, which makes it more similar to human milk and easier for human absorption and utilization. Therefore, goat dairy products, like cheese, yogurt, or butter have superior nutritional value for people than other mammalian milk and dairy products [1,2]. In most developing countries, goat meat has captured the meat consumption market for a long time due to its reasonable prices and its ability to meet the animal protein requirement and nutritional needs of the locals. It is also popular in developed countries [3]. Moreover, cashmere is used as a raw material for high-end clothing due to its great quality. It can be seen that goat products play a key role in people’s lives from the above phenomena, which means increasing goat stocks and improving the goat production performance are particularly important and meaningful. However, the global goat product supplies cannot satisfy people’s higher demand in terms of quantity and quality at present [4]. Hence, it is imperative to improve the reproductive traits and productivity of goats to better solve the imbalance between supply and demand of goat products fundamentally and stimulate the development of the goat industry. 

However, reproductive traits are determined by complex regulatory factors, including the genetic material, nutrient levels, feeding environment, and management, etc. Genetic factors are more stable and easier to select than the more volatile nutrition and environmental factors in regulating goats’ fertility process [5,6,7]. The last decade has witnessed the development of the molecular genetic breeding method in livestock, such as C657T and G749C mutations in the Pregnancy-Associated Glycoprotein gene family (*PAGs*), which has significantly increased the number of live-born piglets (*p* ≤ 0.05) in Hirschmann hybrid-line sows [8]. Genome-wide association analysis and gene set enrichment analysis identified 22 quantitative trait loci (QTLs) and nine gene sets that were closely associated with cattle pregnancy establishment [9]. An 18-bp indel in the 5′ untranslated region (5′ UTR) of pig SRY-box transcription factor 9 (*SOX9*) gene was significantly associated with boar testis weight (*p* < 0.05) and controlled the expression of *SOX9* [10]. Additionally, a single nucleotide polymorphism (SNP) locus within Exon3 of the insulin-like growth factor 1 (*IGF-1*) gene significantly affected the semen quality (*p* < 0.05) in Sanjabi rams [11]. 

In goats and sheep, many high prolific candidate genes have been identified, like *bone morphogenetic protein receptor type 1B* (*BMPR1B*) gene, *bone morphogenetic protein 15* (*BMP15*) gene, *growth differentiation factor 9* (*GDF9*) gene, etc. Meanwhile, some key mutant loci within these genes had been used as useful DNA markers for selecting high prolificacy individuals [12]. The *BMPR1B* gene, also known as the *FecB* gene, plays a major role in sheep prolificacy, and several research projects have identified that the mutation Q249R within the *FecB* gene was highly associated with the ovulation rate in many sheep breeds (e.g., Scottish Blackface Merino, Booroola Merino, Garole, Javanese, Mérinos d’ Arles, and Small Tailed Han ewes) around the world [13,14,15,16,17,18]. Conversely, it showed low polymorphism in goats (e.g., Black Bengal, Beetal, Barbari, Malabari, Osmanabadi, Ganjam, Jining Grey, Wendeng Dairy, and Inner Mongolia Cashmere goats) and had no association with reproductive traits [19,20,21,22]. Moreover, FecX^R^, FecX^H^, FecX^I^, FecX^L^, FecX^G^, and FecX^B^ were six noted SNPs within the sheep *BMP15* gene, and the heterozygous nature of these six loci resulted in an increase in the ovulation rate [23,24,25]. However, except for FecX^B^, no other mutations were found in goats [19,20,26].

Apart from the *FecB* and *BMP15* genes, the polymorphisms of the *GDF9* gene seem to play an important role in the reproductive process of both sheep and goats. Recently, Chu et al. (2018) reported that the expression levels of *GDF9* in the ovaries were higher in high prolific Small Tail Han sheep than that in low prolific specimens, which indicated that *GDF9* may play a positive regulator role in the lambing performance [26]. Furthermore, there are five certain SNP loci—including G1 [27,28], G8 [29], FecTT [30], FecG^E^ [31], and G1111A [32,33]—within *GDF9* associated with the sheep ovulation rate and fecundity. Pan et al. (2015) collected ovary tissues from the multiparous and uniparous goat breeds to analyze the expression characteristics of the *GDF9* gene. However, their study showed no significant difference [34]. It also found that the *GDF9* expression levels of the small antral follicles were nearly 2.5-fold more than those of large antral follicles in Black Bengal and Sirohi goats (two high prolificacy breeds) [35]. Moreover, *GDF9*, except for having the highest expression in the goat’s ovary, is also expressed in 19 other tissues, including the uterus, hypothalamus, pituitary, and more. [34]. All of these studies have demonstrated that the *GDF9* gene is closely related to the differentiation and development of follicles and it may play a wide role in a variety of complex biochemical and physiological processes. However, research studying the mechanism of the goat *GDF9* gene in regulating various normal physiological and metabolic processes of ovaries and how to maintain the normal function of ovaries are still scarce, and further studies are required to clarify these issues. Nevertheless, the genetic polymorphisms of the *GDF9* gene in goats were much richer than in sheep, as well as more complicated. According to the previous literature and the NCBI SNP database, a total of 45 SNP mutations were shown in the goat *GDF9* gene and 15 of them were detected and identified in more than 30 goat breeds around the world (Appendix A, Table 1 and Table 2). An intriguing finding is that each of these 15 SNPs showed a significant association with the goat litter size trait. However, in different research studies, the same SNP mutation exhibited inconsistent characteristics, such as different positive allele and mutation frequencies, as well as diverse effects on lambing. This made it more difficult for the goat breeders to find effective, applicable, and valuable SNPs. Hence, the aim of this study was to summarize and sort the goat *GDF9* SNP loci, and mainly focus on some controversial and potentially effective SNPs with difference effects on goat reproductive traits, in order to find important polymorphic mutations and provide a more effective tool for follow-up research on the genetic selection of goat reproductive traits.

## 2. Mutations in the Goat *GDF9* Gene 

As a member of the transforming growth factor beta (*TGF-β*) superfamily, the *GDF9* gene was first found in the mouse ovaries by McPherron and Lee in 1993 [36]. It was specifically secreted by follicles and acted as a paracrine factor with the BMP15 protein in regulating the proliferation and differentiation of the granulosa cells [37,38,39], and could make the cumulus cells expand by controlling the key enzymes, which suggests the essential role of the *GDF9* gene in the process of folliculogenesis in female mammals [23,40,41]. Moreover, the *GDF9* gene could also stimulate or reduce the expression level of its receptor genes, such as the FSH receptor (*FSHR*) and LH receptor (*LHR*), for the acquisition of oocytes against atresia and developmental competence [40].

In livestock species, the expression of the *GDF9* gene was shown to be specific at the primordial follicle stage within ovine and bovine ovaries [42]. In goats, however, *GDF9* mRNA was expressed in all developmental phases of follicles and was significantly richer in small follicles than in large antral follicles [35,43]. Meanwhile, *GDF9* was expressed broadly in 20 tissues in goats, including the ovary, hypothalamus, pituitary, uterus, and more. The highest expression was exhibited in the ovary, which confirmed that it might play a preponderant role in reproductive organs, and also have wide effects in other organs and tissues [34]. Furthermore, *GDF9* and *FSH* could maintain follicular integrity and promote the activation of primordial follicles and growth during the long-term in vitro culture of goat preantral follicles [44]. These studies further suggested that the *GDF9* gene could plausibly affect goat fecundity through its highest expression in the ovary and interaction with other fecundity genes in different tissues. Therefore, it is of great significance to explore and select some important SNP markers in the goat *GDF9* gene, which like using the Q249R mutation of *FecB* as a key marker gene to improve the fecundity of sheep.

Until now, a total of 15 *GDF9* SNPs have been reported in various goat breeds around the world. However, different studies have shown that the same mutation site has different physiological effects. For instance, the association between one SNP locus and reproductive traits varied with different goat breeds. At the same time, the same allele exhibited positive correlation with high prolificacy in some goat breeds but showed a negative correlation in other goat breeds. These results make it difficult to find a real and effective SNP locus for goat molecule breeding. Therefore, based on the depth and width of previous studies, we selected these 15 controversial and potentially relevant valuable SNPs, and divided them into the following four parts for systematic and comprehensive summary and discussion.

### 2.1. Non-Synonymous SNPs within the Goat GDF9 Gene

#### 2.1.1. V397I Mutation 

V397I, also written as g.4135G>A or p.Val397Ile, is a well-known non-synonymous site within Exon2 of the goat *GDF9* gene and is a benign mutation that does not damage the protein structure [45]. Currently, it has been verified in a total of 32 goat breeds across the world by PCR-SSCP, PCR-RFLP, and sequencing methods. Markhoz goats, which are an endangered Iran breed with the characteristic of variation in the litter size, were used to detect the mutation V397I in 164 individuals via the PCR-RFLP method. These samples were fed in the Sanandaj Markhoz goat Performance Testing Station, located in the Kurdistan province of Iran. No selection on reproduction traits was performed in previous years. The genotyping result showed that the frequency of the mutant allele “A” was 0.686. However, it showed no association with the litter size trait [46]. Ahlawat et al. (2015, 2016) and Maitra et al. (2016) used seven Indian goat breeds, including Black Bengal goats (n = 110/158/158, each number represents the sample size of the each cited research. The following expression has the same meaning), Barbari goats (n = 49/50/0), Ganjam goats (n = 50/50/0), Osmanabadi goats (n = 41/41/0), Beetal goats (n = 28/28/0), Jhakrana goats (n = 13/14/0), and Sangamneri goats (n =50/50/0) (Table 1) to identify the V397I mutation using the PCR-RFLP method [23,47,48]. Although the V397I mutation was detected in the above seven breeds from three studies, the mutation frequency “A” was almost 0.1–0.2 (Table 1). Furthermore, none of these three studies found the mutation V397I had significant effects on litter size, while, in the prolific Black Bengal goats, Ahlawat et al. (2016) found the effect of parity on litter size was significant due to the V397I mutation [47]. This may be due to the small sample size and low individual selection intensity in the above seven goat breeds. 

For Chinese goat breeds, V397I was detected in Shaanbei white cashmere goats (n = 1511) [45], Xinong Saanen dairy goats (n = 241), Guanzhong dairy goats (n = 197), Boer goats (n = 203) [49], Jining Grey goats (n = 178), Guizhou White goats (n = 71), Boer goats (n = 47), and Liaoning cashmere goats (n = 40) [50], as well as in Inner Mongolia cashmere goats (n = 761) [51]. Zhu et al. (2013) and Wang et al. (2013) also found V397I in Big foot black goats (n = 96) and Jintang black goats (n = 81) by PCR-SSCP, and in Henan dairy goats (n = 168), Yaoshan goats (n = 98), and Taihang Black goats (n = 102) by PCR-RFLP [52,53]. Moreover, V397I was also detected in Wendeng goats (n = 40), Beijing native goats (n = 31), Anhui White goats (n = 67), Haimen goats (n = 113), Xuhuai goats (n = 88), Laiwu black goats (n = 32), Laoshan goats (n = 55), Chongming goats (n = 75), Lubei goats (n = 90), and Yimeng goats (n = 80) by using PCR-SSCP and sequencing methods [54,55,56,57,58,59]. In these studies, the frequency of mutant allele “A” was higher in Xinong Saanen dairy goats, Guanzhong dairy goats, Jintang black goats, and Shaanbei white cashmere goats, with a range of nearly 0.45-0.71 (Table 1). In these breeds, the V397I mutation showed significant effects on the litter size trait. However, in other breeds, the “A” frequency was about 0.2 and showed no association with reproductive traits (Table 1), which was similar to the studies in the Indian goat population. This is presumably due to the number of effective mutant populations being small. 

In addition, it is worth noting that the previous studies found V397I in a total of 23 Chinese goat breeds and demonstrated that it was significantly associated with the litter size trait in some breeds. However, the major allele with high prolificacy was not consistent. As seen in Table 3, the major allele with a high prolificacy was the “G” allele (reference allele) in Jining Grey, Inner Mongolia cashmere, and Laiwu black goats. The “GG” genotype have greater litter size than the “AA” genotype (*p* < 0.05), which suggests this SNP mutation played negative effects on litter size traits in these three breeds. However, in Xinong Saanen dairy, Guanzhong dairy, Boer, Anhui White, Big foot black, and Jintang black goats, the litter size of individuals with the “AA” genotype was better than that of individuals with the “GG” genotype (*p* < 0.05). It can be concluded that the V397I mutation showed positive effects on the litter size in these six goat breeds. Furthermore, the V397I mutation exhibited heterosis in Shaanbei white cashmere goats. The individuals with the “GA” genotype had the greatest litter size (*P* = 0.015). Recently, Mahmoudi et al. (2019) used meta-analysis to investigate effects of V397I on litter size, which used four different genetic models to calculate the pooling results from parts of the above published studies, and it was found that the effects were inconsistent in different models but the sensitivity analysis suggested that this SNP negatively affected the goat litter size trait [60]. These results suggested that V397I seems to play a “true” role in goat reproductive traits, but the inconsistent results might be due to the fact that this mutation is a minor quantitative trait nucleotide (QTN) that is linked with other stabilized QTNs during the long-term selection of evolution in goat breeds with different economic performances.

#### 2.1.2. Q320P Mutation 

The Q320P mutation is an A to C transversion at the 3905nt within Exon2 of the goat *GDF9* gene (written as g.3905A>C or p.Gln320Pro). Zhao et al. (2016) pointed out that the “AA” and the “CC” genotypes were differentially distributed (*p* < 0.05) in high-fecundity (more than two litters for a continuous six years) and low-fecundity Inner Mongolia cashmere goats via sequencing [51]. Moreover, Q320P was detected in Yangtse River Delta White goats (n = 105), Huanghuai goats (n = 40), and Boer goats (n = 55) by PCR-SSCP and sequencing methods [61], as well as in Jining Grey goats (n = 234), Lubei White goats (n = 90), and Yimeng Black goats (n = 80) [58]. By using PCR-RFLP and sequencing methods, the Q320P mutation was also identified in Shaanbei white cashmere goats (n = 1511) [45], Jining Grey goats (n = 177), Guizhou White goats (n = 71), Boer goats (n = 47), and Liaoning cashmere goats (n = 41) [50]. The frequency of the mutant allele “C” in the above breeds was about 0.15-0.68 (Table 1). Moreover, among these results, only Wang et al. (2019) and Feng et al. (2011) found that the Q320P mutation had significant positive effects on litter size in Shaanbei white cashmere goats (low fecundity breed) and Jining Grey goats (high fecundity breed) (*p* < 0.5) with an accordant dominant genotype “CC.” The frequencies of the mutant allele “C” were 0.286 and 0.350, respectively (Table 4) [45,50]. 

By contrast, Arefnejad et al. (2018) found that Q320P significantly affected the litter size in a negative way with a reference allele “A” in 120 Markhoz goats [46]. However, Shokrollahi and Morammazi (2018) recently adopted a larger sample size of Markhoz goats (n = 146) for research and found that the frequency of the mutant allele was similar (about 0.6), but it did not affect the litter size trait anymore (Table 4) [62]. This suggests that the sample size might be a factor that has influenced the conclusions made by the researchers. Larger experimental populations are needed for further study to improve the reliability and accuracy of the results. In addition, Ahlawat et al. (2015,2016) found Q320P polymorphisms in five Indian native goat breeds, including Black Bengal goats (n = 158/110), Barbari goats (n = 0/49), Ganjam goats (n = 0/50), Osmanabadi goats (n = 0/41), and Sangamneri goats (n = 0/50) by the PCR-RFLP method [23,47]. Maitra et al. (2016) also found this mutation in the Indian goat breeds Beetal (n = 28) and Jhakrana (n = 14), but not in the above five breeds [48]. However, Q320P showed no effects on the litter size trait of any of the above seven Indian breeds, which might due to the low mutation frequency (no greater than 5%). Therefore, based on the results of studies in Chinese, Iranian, and Indian goats, the effect of Q320P locus on the litter size trait was unstable and varied with the change of goat breeds. Nevertheless, it showed great and stable effects on Chinese cashmere goat reproductive traits with a large sample size (n = 1511). Therefore, it is of great significance to find the true role of Q320P via further research with larger sample sizes and a wide range of global goat breeds, as well as the mechanism of Q320P in the reproductive traits.

#### 2.1.3. A240V Mutation

Dong and Du first found A240V (also known as g.3665C>T and p.Ala240Val) in Jining Grey goats (n = 234), Lubei White goats (n = 90), and Yimeng Black goats (n = 80) [58]. The minor allele frequency (MAF) of the mutant allele “T” was 0.092, 0.009, and 0.116, respectively, and there was no significant association between A240V and the litter size trait of these three breeds. Subsequently, Wang et al. (2013) used Henan dairy goats (n = 116), Yaoshan goats (n = 98), and Taihang Black goats (n = 102) to further explore whether the A240V has an influence on goat reproduction. Interestingly, only “CC” and “CT” genotypes were identified and the frequency of the “T” allele was still low, with values of 0.075, 0.036, and 0.024, respectively. Nonetheless, the association analysis showed the A240V significantly affected the first-born litter size of Henan dairy goats (*p* < 0.05), which heterozygous individuals (genotype CT) had 0.60 kids more than those of the wild homozygote individuals (genotype CC) [53]. In the previously mentioned studies, there were no mutant homozygotes in six goat breeds and the mutation frequency was lower, which might be due to the elimination of “TT” individuals from goat breeding over time. Furthermore, the “CT” genotype showed greater litter size but only in one breed. Therefore, it is necessary to focus on more breeds and a large experimental population to further study the relationship between the A240V and goat reproduction.

### 2.2. Synonymous SNPs within the Goat GDF9 Gene

#### 2.2.1. L61L Mutation

Non-synonymous mutations could lead to the replacement of encoded amino acids and change the protein structure and functions. Apart from it, synonymous mutations could not change the composition of the peptide chain directly. However, it is still important in the translation process, which could alter the programmed translational velocity and then influence the encoded protein folding and function [63]. Therefore, the synonymous SNPs of the goat *GDF9* gene could not be ignored because they may also have key effects on goat fecundity. Synonymous mutation L61L (also known as g.2006C>A or p.Leu61Leu ) was located at Exon1 of the goat *GDF9* gene. Chu et al. (2011) found this SNP in Jining Grey goats (n = 224), Wendeng dairy goats (n = 40), Liaoning cashmere goats (n = 39), Beijing native goats (n = 40), and Boer goats (n = 39) by PCR-SSCP and sequencing methods. The results showed that the MAF of the “C” allele (reference allele) was between 0.09 and 0.39, which suggested that the mutation frequency was higher in these goat breeds (Table 2). Meanwhile, association analysis indicated that L61L showed a significant negative effect on the litter size of the prolific breed Jining Grey goats, and mutant homozygote “AA” had the lowest litter size (*p* < 0.05) [64]. Moreover, L61L was detected in Anhui White goats (n = 68), Lubei goats (n = 90), and Yimeng goats (n = 80), and it also affected the litter size of Anhui White goats and Yimeng goats, for which the dominant genotypes with a high fecundity were consistent with the study by Chu et al. (2011) [54,58]. It can be seen that there was a negative correlation between the L61L and the goat litter size trait, which means it could be used as a marker to select the high fecundity individuals in the goat breeding process. Furthermore, it is meaningful to further explore the effect of the L61L mutation on reproductive traits in other goat breeds. 

#### 2.2.2. N121N Mutation 

Chu et al. (2011) firstly reported synonymous N121N in the goat *GDF9* gene by PCR-SSCP and sequencing methods, which is also written as g.2159C>T and p.Asn336Asn. The MAF of the “T” allele (mutant allele) was about 0.12 in low prolificacy breeds, such as Wendeng dairy goats (n = 40), Liaoning cashmere goats (n = 39), Beijing native goats (n = 40), and Boer goats (n = 39) [64]. However, in high prolificacy among Jining Grey goats (n = 224), it reached 0.39, whose distribution was different (*p* < 0.01) (Table 2) [64]. Meanwhile, the correlation analysis results showed N121N had a significant positive effect on the first-born litter size of Jining Grey goats (*p* < 0.05), and the mutant homozygote “TT” had 0.72 kids more than the wild type. However, it had no effects on the other goat breeds [58]. Therefore, it can be concluded that this mutation has higher mutant frequency in high prolificacy goat breeds (*p* < 0.01) and mutation on this locus could significantly improve the reproductive traits of the high prolificacy goat breed, but it also needs to be further explored in more high or low prolificacy goat breeds to make this conclusion more accurate.

#### 2.2.3. L141L Mutation 

L141L, which is also known as g.3369G>A and p.Leu423Leu, is the only synonymous mutation verified within the Exon 2 of goat *GDF9* gene. It was found in Jining Grey goats (n = 178/109/60), Guizhou White goats (n = 71), Boer goats (n = 47/28), Liaoning cashmere goats (n = 40/38), Wendeng dairy goats (n = 40), Beijing native goats (n = 31), Laiwu Black goats (n = 32), and Laoshan dairy goats (n = 55) by PCR-SSCP and sequencing [47,56,59]. In Laoshan dairy and Wendeng dairy goats, the frequencies of the mutant allele “A” was 0.709 and 0.687, respectively. However, in other goat breeds, this mutation rate was only about 15% (Table 2), which suggests that L141L might be a selected marker subject to intense selection during the breeding of dairy goats. Moreover, this mutation was associated with the first-born litter size trait in the high fertility breed Jining Grey goats, in which the individuals with “GA” and “AA” genotypes showed nearly 0.6 kids more than those of the individuals with the “GG” genotype (*p* < 0.05). It showed no effects on other low fertility goats, which was possibly due to the breed-specific effect or the limited sample size [51,56,59].

### 2.3. Mutations in the Regulatory Region of the Goat GDF9 Gene

As we know, the mutations in the gene regulatory region generally affect the expression and functions of genes. For example, g.224A>G and g.227C>T within the 5′ flanking region of the *IGF-1* gene were significantly associated with goat litter size [65]. A 14-base pair indel locus in the core promoter region of goat CKLF-like MARVEL transmembrane domain containing 2 (*CMTM2*) gene could significantly decrease the goat litter size [66]. Three novel indel mutations within the promoter and introns of the goat membrane-associated ring finger (C3HC4) 1 (*MARCH1*) gene were associated with the first-born litter size by impacting the expression levels of *MARCH1* [67]. The g.173057T>C within the 3′ UTR of the goat prolactin receptor (*PRLR*) gene had significant effects on litter size by altering the binding site of bta-miR-302a [68]. The 10 base pair indel located in the Intron19 of goat sperm flagella 2 (*SPEF2*) gene, which is a fragment binding to the androgen receptor (AR), was significantly correlated with litter size [69]. It is worth noting that, based on the NCBI SNP database, we found that half of the mutations were located in the regulatory region of the goat *GDF9* gene, which includes Intron1, promoter, 3′ UTR, and 3′ flanking. However, only two loci were identified and one was g.3288G>A, which was located in Intron1 and it had been detected in the Jining Grey goat (n = 175), Guizhou White goat (n = 71), Boer goat (n = 47), and Liaoning Cashmere goat (n = 40) with a mutant frequency of 0.06, 0.01, 0.19, and 0.34, respectively. Nevertheless, this mutation had no effects on the litter size (Appendix A) [50]. The other was a 12-base pair indel (NC_022299.1:g.66028950insTACTTTCAACAA, rs670709574) in the 3′ regulatory region, which was detected in 1328 Shaanbei white cashmere goats. Its mutant frequency was 0.455. In addition, this indel could affect the first-born litter size, but is due to the main effects of goat growth traits [70]. Therefore, neither have a direct effect on the goat reproductive traits. Whether unverified loci could play a role in goat reproductive traits need further research. 

### 2.4. Other Mutations within the Goat GDF9 Gene

In addition to the previously reported SNP mutations, there were also many tested SNP mutations that have been found to be unrelated to reproductive traits in goats and untested mutations. For example, G40G (g.1941C>T/p.Gly40Gly) and N112N (g.2159C>T/p.Asn112Asn) were two synonymous SNP loci within the goat *GDF9* gene, which have been identified in different Chinese goat breeds. Yang et al. (2012) found G40G in two local Chinese goat breeds with opposite fecundity, which include Lezhi black goats (n = 6) and Tibetan goats (n = 6), by using RT-PCR and sequencing methods [71]. Chu et al. (2011) found N112N in four Chinese indigenous goat breeds and Boer goats, but it did not affect the litter size trait [64]. Moreover, D129D (g.2211G>A/p.Asp129Asp) and S165S (g.3441C>A/p.Ser165Ser) were detected in Indian Assam hill goats by PCR-RFLP, and both of them showed no association with litter size [72]. For non-synonymous SNP mutations, L50P (g.1972T>C/p.Leu50Pro) and A273V (g.3764C>T/p.Ala273Val) were two loci that have received less attention and study. L50P was detected in six Lezhi black goats (prolific breed) and six Tibetan goats (non-prolific breed) by using RT-PCR and sequencing [71]. A273V was mainly exhibited in seven Indian goat breeds and had no association with prolificacy traits [23,47,48,73]. Furthermore, the mutations P27R (g.1902C>G/p.Pro27Arg), A85G (g.2077C>G/p.Ala85Gly), E204Q (g.3556G>C/p.Glu204Gln), and T217T (g.3597G>T/p.Thr217Thr) were searched for in the NCBI SNP database but have not been identified and reported in any goat breeds so far. This part of the SNP-related research is relatively lacking, and there is still a need to further explore and research a rich variety of goats.

## 3. Conclusions

In this study, we gathered and sorted 45 SNP loci from the goat *GDF9* gene, and mainly analyzed and discussed the relationship between part potentially “true” SNPs and the goat reproductive traits. Among these mutations, three non-synonymous mutations, which include A240V, Q320P, and V397I, and three synonymous mutations, which include L61L, N121N, and L141L, were found to have a high mutant frequency in many fecundity goat breeds, especially the Q320P, V397I, L61L, and N112N, which are as high as 0.5–0.7 (Table 1 and Table 2). Meanwhile, these six SNPs significantly related to the litter size trait in Chinese goat breeds, while some results of these studies were not consistent. For example, the major allele with high prolificacy of the V397I mutation was the mutant allele “G” in Shaanbei white cashmere, Inner Mongolia cashmere, Lubei White, Jining Grey, and Laiwu black goats [51,56,58], but the reference allele “A” was found with greater litter size in Xinong Saanen dairy, Guanzhong dairy, Big foot black, Jintang black, and Yimeng black goats [29,52,58]. This is likely due to the breed-specific effect or the effects of this locus was due to a linkage with other mutations [45,51]. Therefore, it would be meaningful to detect these six SNPs in more goat breeds or in large herds and conduct further functional gain or loss experiments to explore the influencing mechanism and explain this theory more clearly. Furthermore, it is noteworthy that the previous studies mostly focused on a single SNP site and ignored the fact that reproductive traits are complex quantitative traits involving QTL, QTN, and interactions by the fecundity gene. Only Wang et al. (2019) found that Q320P and V397I mutations were well linked in cashmere goats, which needs to be further explored in other goat breeds around the world [45]. Therefore, it is necessary to pay attention to the combined effects of multiple loci on goat reproductive traits. 

According to the summary and analytic results of the current SNP loci within the goat *GDF9* gene in this study, it was found that A240V, Q320P, V397I, L61L, N121N, and L141L are six effective SNPs associated with the litter size trait. In most goat breeds worldwide, the V397I and L61L mutations showed a negative relationship with strong goat fertility and the other four SNPs exhibited a positive effect. These findings provide important information to develop goat molecular breeding strategies and improve the reproduction and production performance of goats.

## Figures and Tables

**Table 1 animals-09-00886-t001:** All non-synonymous SNP loci (verified and predicted) within the goat *GDF9* gene.

SNP Locus	Rs of the SNP	Breed and Sample Size	Mutant allele Frequency	Effect of the Mutant Allele on the Litter Size Trait	*p* Values	Reference
g.1972T>C/ c.149T>C/p.L50P	-	Lezhi black goats, n = 6	-	-	-	[70]
	-	Tibetan goats, n = 6	-	-	-	
g.3665C>T/ c.719C>T/ p.A240V	rs637835524	Henan dairy goats, n = 166	0.075	Positive	*p* < 0.05	[53]
		Yaoshan goats, n = 98	0.036	-	-	
		Taihang Black goats, n = 102	0.024	-	-	
g.3665C>T/ c.719C>T/ p.A240V	rs637835524	Jining Grey goats, n = 234	0.092	-	*p* > 0.05	[58]
		Lubei White goats, n = 90	0	-	*p* > 0.05	
		Yimeng Black goats, n = 80	0.116	-	*p* > 0.05	
g.3764C>T/ c.818C>T/ p.A273V	rs662668357	Black Bengal goats, n = 158	026	-	*p* > 0.05	[23]
g.3764C>T/ c.818C>T/ p.A273V	rs662668357	Black Bengal goats, n = 110	0.34	-	*p* > 0.05	[47]
		Barbari, n = 49	0.082	-	*p* > 0.05	
		Beetal, n = 28	0	-	*p* > 0.05	
		Ganjam, n = 50	0.04	-	*p* > 0.05	
		Jhakrana, n = 14	0	-	*p* > 0.05	
		Osmanabadi, n = 41	0	-	*p* > 0.05	
		Sangamneri, n = 50	0.04	-	*p* > 0.05	
g.3764C>T/ c.818C>T/ p.A273V	rs662668357	Black Bengal goats, n = 158	0.256	-	-	[48]
		Barbari, n = 50	0.06	-	-	
		Beetal, n = 28	0	-	-	
		Ganjam, n = 50	0.02	-	-	
		Jhakrana, n = 14	0	-	-	
		Osmanabadi, 41	0	-	-	
		Sangamneri, n = 50	0.02	-	-	
g.3905A>C/ c.959A>C/ p.Q320P	rs645345606	Markhoz goats, n = 120	0.623	Positive	*p* < 0.05	[61]
g.3905A>C/ c.959A>C/ p.Q320P	rs645345606	Markhoz goats, n = 164	0.686	-	*p* > 0.05	[46]
g.3905A>C/ c.959A>C/ p.Q320P	rs645345606	Black Bengal goats, n = 110	0.032	-	*p* > 0.05	[47]
		Barbari, n = 49	0.041	-	*p* > 0.05	
		Beetal, n = 27	0	-	*p* > 0.05	
		Ganjam, n = 50	0.04	-	*p* > 0.05	
		Jhakrana, n = 13	0	-	*p* > 0.05	
		Osmanabadi, n = 41	0.024	-	*p* > 0.05	
		Sangamneri, n = 50	0.07	-	*p* > 0.05	
g.3905A>C/ c.959A>C/ p.Q320P	rs645345606	Black Bengal goats, n = 158	0.035	-	-	[48]
		Barbari, n = 50	0.040	-	-	
		Beetal, n = 28	0.018	-	-	
		Ganjam, n = 50	0.030	-	-	
		Jhakrana, n = 14	0.036	-	-	
		Osmanabadi, n = 41	0.024	-	-	
		Sangamneri, n = 50	0.070	-	-	
g.3905A>C/ c.959A>C/ p.Q320P	rs645345606	Jining Grey goats, n = 177	0.350	Positive	*p* < 0.05	[50]
		Guizhou White goats, n = 71	0.373	-	*p* > 0.05	
		Boer goats, n = 47	0.70	-	*p* > 0.05	
		Liaoning cashmere goats, n = 40	0.200	-	*p* > 0.05	
g.3905A>C/ c.959A>C/ p.Q320P	rs645345606	Black Bengal goats, n = 158	0.040	-	*p* > 0.05	[23]
g.3905A>C/ c.959A>C/ p.Q320P	rs645345606	Lubei White goats, n = 90	0151	-	*p* > 0.05	[58]
		Yimeng Black goats, n = 80	0.182	-	*p* > 0.05	
g.3905A>C/ c.959A>C/ p.Q320P	rs645345606	Yangtse River Delta White goats, n = 105	0.262	-	-	[60]
		Huanghuai goats, n = 40	0.150	-	-	
		Boer goats, n = 35	0.086	-	-	
g.3905A>C/ c.959A>C/ p.Q320P	rs645345606	Shaanbei white cashmere goats, n = 1511	0.286	Positive	*p* < 0.05	[45]
g.4135A>G/c.1189A>G/ p.V397I	rs637044681	Shaanbei white cashmere goats, n = 1511	0.523	Negative	*p* < 0.05	[45]
g.4135A>G/c.1189A>G/ p.V397I	rs637044681	Markhoz goats, n = 164	0.314	-	*p* > 0.05	[46]
g.4135A>G/c.1189A>G/ p.V397I	rs637044681	Xinong Saanen dairy goats, n = 241	0.290	Positive	*p* < 0.05	[49]
		Guanzhong dairy goats, n = 197	0290	Positive	*p* < 0.05	
		Boer goats, n = 203	0.290	Positive	*p* < 0.05	
g.4135A>G/c.1189A>G/ p.V397I	rs637044681	Jining Grey goats, n = 178	0.83	-	*p* > 0.05	[50]
		Guizhou White goats, n = 71	0.96	-	-	
Continued in Table 1						
		Boer goats, n = 47	0.81	-	-	
		Liaoning cashmere goats, n = 40	0.64	-	-	
g.4135A>G/c.1189A>G/ p.V397I	rs637044681	Henan dairy goats, n = 168	0.83	Positive	*p* < 0.05	[53]
		Yaoshan goats, n = 98	0.92	-	-	
		Taihang Black goats, n = 102	0.89	-	-	
g.4135A>G/c.1189A>G/ p.V397I	rs637044681	Black Bengal goats, n = 110	0.92	-	*p* > 0.05	[47]
		Barbari, n = 49	0.90	-	*p* > 0.05	
		Beetal, n = 28	0.64	-	*p* > 0.05	
		Ganjam, n = 50	0.93	-	*p* > 0.05	
		Jhakrana, n = 13	0.50	-	*p* > 0.05	
		Osmanabadi, n = 41	0.99	-	*p* > 0.05	
		Sangamneri, n = 50	0.78	-	*p* > 0.05	
g.4135A>G/c.1189A>G/ p.V397I	rs637044681	Black Bengal goats, n = 158	0.89	-	-	[48]
		Barbari, n = 50	0.90	-	-	
		Beetal, n = 28	0.61	-	-	
		Ganjam, n = 50	0.93	-	-	
		Jhakrana, n = 14	0.54	-	-	
		Osmanabadi, n = 41	0.84	-	-	
		Sangamneri, n = 50	0.78	-	-	
g.4135A>G/c.1189A>G/ p.V397I	rs637044681	Inner Mongolia cashmere goats, n = 761	0.68	Positive	*p* < 0.05	[51]
g.4135A>G/c.1189A>G/ p.V397I	rs637044681	Big foot black goats, n = 96	0.57	Negative	*p* < 0.05	[52]
		Jintang black goats, n = 81	0.32	Negative	*p* < 0.05	
g.4135A>G/c.1189A>G/ p.V397I	rs637044681	Black Bengal goats, n = 158	0.89	-	*p* > 0.05	[23]
g.4135A>G/c.1189A>G/ p.V397I	rs637044681	Jining Grey goats, n = 109	0.93	Positive	*p* < 0.05	[59]
		Wendeng dairy goats, n = 40	0.31	-	-	
		Liaoning cashmere goats, n = 38	0.40	-	-	
		Beijing native goats, n = 31	0.68	-	-	
		Boer goats, n = 28	0.57	-	-	
g.4135A>G/c.1189A>G/ p.V397I	rs637044681	Anhui White goats, n = 67	0.72	Negative	*p* < 0.05	[54]
g.4135A>G/c.1189A>G/ p.V397I	rs637044681	Haimen goats, n = 113	0.77	-	*p* > 0.05	[55]
		Xuhuai goats, n = 88	0.89	-	*p* > 0.05	
g.4135A>G/c.1189A>G/ p.V397I	rs637044681	Laiwu Black goats, n = 32	0.86	Positive	*p* < 0.05	[56]
		Jining Grey goats, n = 60	0.92	Positive	*p* < 0.05	
		Laoshan dairy goats, n = 55	0.40	-	*p* > 0.05	
g.4135A>G/c.1189A>G/ p.V397I	rs637044681	Chongming White goats, n = 75	0.80	-	*P* >0.05	[57]
g.4135A>G/c.1189A>G/ p.V397I	rs637044681	Jining Grey goats, n = 234	0.80	-	*p* > 0.05	[58]
		Lubei White goats, n = 90	0.68	Negative	*p* < 0.05	
		Yimeng Black goats, n = 80	0.85	Negative	*p* < 0.05	
g.1902C>G/c.79C>G/ p.P27R	rs671913497	-	-	-	-	-
g.2077C>G/c.254C>G/ p.A85G	rs654628150	-	-	-	-	-
g.3556G>C/ c.610G>C/ p.E204Q	rs666975374	-	-	-	-	-

**Table 2 animals-09-00886-t002:** All synonymous SNP loci (verified and predicted) within goat *GDF9* gene.

SNP Locus	Rs of the SNP	Breed and Sample Size	Mutant Allele Frequency	Effect of the Mutant Allele on the Litter Size Trait	*p* Values	Reference
g.1941C>T/ c.118C>T/p.G40G	-	Lezhi black goats, n = 6	-	-	-	[70]
	-	Tibetan goats, n = 6	-	-	-	
g.2006C>A/ c.183C>A/p.L61L	rs669811820	Jining Grey goats, n = 224	0.61	Negative	*P* < 0.05	[63]
		Wendeng dairy goats, n = 40	0.84	-	-	
		Liaoning cashmere goats, n = 39	0.88	-	-	
		Beijing native goats, n = 40	0.91	-	-	
		Boer goats, n = 39	0.88	-	-	
g.2006C>A/ c.183C>A/p.L61L	rs669811820	Jining Grey goats, n = 234	0.33	-	*P* > 0.05	[58]
		Lubei White goats, n = 90	0.20	-	*P* > 0.05	
		Yimeng Black goats, n = 80	0.35	Negative	*P* < 0.05	
g.2006C>A/ c.183C>A/p.L61L	rs669811820	Anhui White goats, n = 68	0.618	Negative	*P* < 0.05	[54]
g.2159C>T/c.336C>T/p.N112N	-	Jining Grey goats, n = 224	0.39	Positive	*P* < 0.05	[63]
		Wendeng dairy goats, n = 40	0.16	-	-	
		Liaoning cashmere goats, n = 39	0.12	-	-	
		Beijing native goats, n = 40	0.09	-	-	
		Boer goats, n = 39	0.12	-	-	
g.2211G>A/c.387G>A/p.D129D	-	Assam hill goat, n = 92	-	-	-	[71]
g.3441C>A/c.495C>A/p.S165S	-	Assam hill goat, n = 92	-	-	-	[71]
g.3369G>A/ c.423G>A/p.L141L	rs650650729	Jining Grey goats, n = 178	0.104	-	*P* > 0.05	[50]
		Guizhou White goats, n = 71	0.021	-	*P* > 0.05	
		Boer goats, n = 47	0.192	-	*P* > 0.05	
		Liaoning cashmere goats, n = 40	0.363	-	*P* > 0.05	
g.3369G>A/ c.423G>A/p.L141L	rs650650729	Jining Grey goats, n = 109	0.087	Negative	*P* < 0.05	[59]
		Wendeng dairy goats, n = 40	0.313	-	-	
		Liaoning cashmere goats, n = 38	0.605	-	-	
		Beijing native goats, n = 31	0.097	-	-	
		Boer goats, n = 28	0.304	-	-	
g.3369G>A/ c.423G>A/p.L141L	rs650650729	Laiwu Black goats, n = 32	0.187	-	*P* > 0.05	[56]
		Jining Grey goats, n = 60	0.108	Negative	*P* < 0.05	
		Laoshan dairy goats, n = 55	0.709	-	*P* > 0.05	
g.3597G>T/ c.651G>T/p.T217T	rs651511232	-	-	-	-	-

**Table 3 animals-09-00886-t003:** V397I mutation within the *GDF9* gene associated with litter size in global goat breeds.

Number	Test Method	Country	Breeds	Sample Size	Minor Allele Frequency	Parity	*p* Values	Dominant Genotype	Reference
1	PCR-RFLP Sequencing	China	Shaanbei white cashmere goats	1511	A-0.477	1st	*p* < 0.05	AG	
2	PCR-RFLP Sequencing	China	Xinong Saanen dairy goats	241	G-0.290	1st	*p* < 0.05	AA, GA	[49]
	PCR-RFLP Sequencing	China	Xinong Saanen dairy goats	241	G-0.290	2nd	*p* < 0.05	AA	
	PCR-RFLP Sequencing	China	Xinong Saanen dairy goats	241	G-0.290	3rd	*p* < 0.05	AA, GA	
	PCR-RFLP Sequencing	China	Xinong Saanen dairy goats	241	G-0.290	4th	*p* > 0.05	AA	
	PCR-RFLP Sequencing	China	Xinong Saanen dairy goats	241	G-0.290	Average	*p* < 0.05	AA, GA	
	PCR-RFLP Sequencing	China	Guanzhong dairy goats	197	G-0.290	1st	*p* < 0.05		
	PCR-RFLP Sequencing	China	Guanzhong dairy goats	197	G-0.290	2nd	*p* < 0.05	AA	
	PCR-RFLP Sequencing	China	Guanzhong dairy goats	197	G-0.290	3rd	*p* < 0.05	AA, GA	
	PCR-RFLP Sequencing	China	Guanzhong dairy goats	197	G-0.290	4th	*p* > 0.05	AA	
	PCR-RFLP Sequencing	China	Guanzhong dairy goats	197	G-0.290	Average	*p* < 0.05	AA, GA	
	PCR-RFLP Sequencing	China	Boer goats	203	G-0.290	1st	*p* < 0.05		
	PCR-RFLP Sequencing	China	Boer goats	203	G-0.290	2nd	*p* < 0.05	AA	
	PCR-RFLP Sequencing	China	Boer goats	203	G-0.290	3rd	*p* < 0.05	AA, GA	
	PCR-RFLP Sequencing	China	Boer goats	203	G-0.290	4th	*p* > 0.05	AA	
	PCR-RFLP Sequencing	China	Boer goats	203	G-0.290	Average	*p* < 0.05	AA, GA	
3	PCR-RFLP Sequencing	China	Jining Grey goats	178	A-0.171	1st	*p* > 0.05	GG	[50]
	PCR-RFLP Sequencing	China	Guizhou White goats	71	A-0.032				
	PCR-RFLP Sequencing	China	Boer goats	41	A-0.192				
	PCR-RFLP Sequencing	China	Liaoning cashmere goats	40	A-0.363				
4	PCR-RFLP Sequencing	China	Henan dairy goats	168	A-0.173	1st	*p* < 0.05	GA	[53]
	PCR-RFLP Sequencing	China	Yaoshan goats	98	A-0.082				
	PCR-RFLP Sequencing	China	Taihang Black goats	102	A-0.113				
5	PCR-RFLP	India	Black Bengal goats	110	A-0.180	1st	*p* > 0.05	GA	[47]
	PCR-RFLP	India	Black Bengal goats	110	A-0.180	2nd	*p* > 0.05	AA	
	PCR-RFLP	India	Black Bengal goats	110	A-0.180	3rd	*p* > 0.05	AA	
	PCR-RFLP	India	Barbari	49	A-0.102	1st	*p* > 0.05	GA	
	PCR-RFLP	India	Barbari	49	A-0.102	2nd	*p* > 0.05	GA	
	PCR-RFLP	India	Barbari	49	A-0.102	3rd	*p* > 0.05	GA	
	PCR-RFLP	India	Beetal	28	A-0.357	1st	*p* > 0.05	GG	
	PCR-RFLP	India	Beetal	28	A-0.357	2nd	*p* > 0.05	GG	
	PCR-RFLP	India	Beetal	28	A-0.357	3rd	*p* > 0.05	GG	
	PCR-RFLP	India	Ganjam	50	A-0.070	1st	*p* > 0.05	GG	
	PCR-RFLP	India	Ganjam	50	A-0.070	2nd	*p* > 0.05	GG	
	PCR-RFLP	India	Ganjam	50	A-0.070	3rd	*p* > 0.05	GG	
	PCR-RFLP	India	Jhakrana	13	A-0.500	1st	*p* > 0.05		
	PCR-RFLP	India	Jhakrana	13	A-0.500	2nd	*p* > 0.05		
	PCR-RFLP	India	Jhakrana	13	A-0.500	3rd	*p* > 0.05		
	PCR-RFLP	India	Osmanabadi	41	A-0.012	1st	*p* > 0.05	GA	
	PCR-RFLP	India	Osmanabadi	41	A-0.012	2nd	*p* > 0.05	GG	
	PCR-RFLP	India	Osmanabadi	41	A-0.012	3rd	*p* > 0.05	GA	
	PCR-RFLP	India	Sangamneri	50	A-0.220	1st	*p* > 0.05	GA	
	PCR-RFLP	India	Sangamneri	50	A-0.220	2nd	*p* > 0.05	GA	
	PCR-RFLP	India	Sangamneri	50	A-0.220	3rd	*p* > 0.05	GG	
6	PCR-RFLP	India	Black Bengal goats	158	A-0.108				[48]
	PCR-RFLP	India	Barbari	50	A-0.100				
	PCR-RFLP	India	Beetal	28	A-0.393				
	PCR-RFLP	India	Ganjam	50	A-0.070				
	PCR-RFLP	India	Jhakrana	14	A-0.464				
	PCR-RFLP	India	Osmanabadi	41	A-0.159				
	PCR-RFLP	India	Sangamneri	50	A-0.220				
7	PCR-RFLP Sequencing	China	Inner Mongolia cashmere goats	761	A-0.320	1st	*p* < 0.05	GG	[51]
	PCR-RFLP Sequencing	China	Inner Mongolia cashmere goats	761	A-0.320	2nd	*p* < 0.05	GG	
	PCR-RFLP Sequencing	China	Inner Mongolia cashmere goats	761	A-0.320	3rd	*p* > 0.05	GG	
	PCR-RFLP Sequencing	China	Inner Mongolia cashmere goats	761	A-0.320	4th	*p* < 0.05	GG	
	PCR-RFLP Sequencing	China	Inner Mongolia cashmere goats	761	A-0.320	5th	*p* < 0.05	GG	
	PCR-RFLP Sequencing	China	Inner Mongolia cashmere goats	761	A-0.320	6th	*p* < 0.05	GG	
	PCR-RFLP Sequencing	China	Inner Mongolia cashmere goats	761	A-0.320	Average	*p* < 0.05	GG	
8	PCR-SSCP	China	Big foot black goats	96	A-0.430		*p* < 0.05	AA, GA	[52]
	PCR-SSCP	China	Jintang black goats	81	G-0.320		*p* < 0.05	AA, GA	
9	PCR-RFLP	India	Black Bengal goats	158	A-0.110		*p* > 0.05	GA	[23]
10	PCR-SSCP Sequencing	China	Jining Grey goats	109	A-0.073	1st	*p* < 0.05	GG	[59]
	PCR-SSCP Sequencing	China	Wendeng dairy goats	40	G-0.313				
	PCR-SSCP Sequencing	China	Liaoning cashmere goats	38	G-0.395				
	PCR-SSCP Sequencing	China	Beijing native goats	31	A-0.323				
	PCR-SSCP Sequencing	China	Boer goats	28	A-0.429				
11	PCR-SSCP Sequencing	China	Anhui White goats	67	A-0.284	1st	*p* < 0.05	AA	[54]
12	PCR-SSCP Sequencing	China	Haimen goats	113	A-0.230	1st	*p* > 0.05	GA	[55]
	PCR-SSCP Sequencing	China	Xuhuai goats	88	A-0.110	1st	*p* > 0.05		
13	PCR-SSCP Sequencing	China	Laiwu Black goats	32	A-0.141	1st	*p* < 0.05	GG	[56]
	PCR-SSCP Sequencing	China	Jining Grey goats	60	A-0.083	1st	*p* < 0.05	GG	
	PCR-SSCP Sequencing	China	Laoshan dairy goats	55	A-0.600	1st	*p* > 0.05	GA	
14	PCR-SSCP Sequencing	China	Chongming goats	75	A-0.200	1st	*p* > 0.05	GA	[57]
15	PCR-SSCP Sequencing	China	Lubei White goats	90	A-0.325	1st	*p* < 0.05	AG, GG	[58]
	PCR-SSCP Sequencing	China	Yimeng Black goats	80	A-0.148	1st	*p* < 0.05	AA	
16	PCR-RFLP	Iran	Markhoz goats	164	G-0.314	1st	*p* > 0.05		[46]

**Table 4 animals-09-00886-t004:** Q320*P* mutation within the *GDF9* gene associated with litter size in global goat breeds.

Number	Country	Breeds	Sample Size	Minor Allele Frequency	Parity	*p* Values	Dominant Genotype	Reference
1	China	Shaanbei white cashmere goats	1511	C-0.286	1st	*p* < 0.05	AC, CC	[45]
2	India	Black Bengal goats	110	C-0.032	1st	*p* > 0.05	AC	[47]
	India	Black Bengal goats	110	C-0.032	2nd	*p* > 0.05	AA	
	India	Black Bengal goats	110	C-0.032	3rd	*p* > 0.05	AC	
	India	Barbari	49	C-0.041	1st	*p* > 0.05	AA	
	India	Barbari	49	C-0.041	2nd	*p* > 0.05	AA	
	India	Barbari	49	C-0.041	3rd	*p* > 0.05	AA	
	India	Beetal	27	C-0	1st	*p* > 0.05	AA	
	India	Beetal	27	C-0	2nd	*p* > 0.05	AA	
	India	Beetal	27	C-0	3rd	*p* > 0.05	AA	
	India	Ganjam	46	C-0.040	1st	*p* > 0.05	AA	
	India	Ganjam	46	C-0.040	2nd	*p* > 0.05	AA	
	India	Ganjam	46	C-0.040	3rd	*p* > 0.05	AA	
	India	Jhakrana	13	C-0	1st	*p* > 0.05		
	India	Jhakrana	13	C-0	2nd	*p* > 0.05		
	India	Jhakrana	13	C-0	3rd	*p* > 0.05		
	India	Osmanabadi	39	C-0.024	1st	*p* > 0.05	AA	
	India	Osmanabadi	39	C-0.024	2nd	*p* > 0.05	AA	
	India	Osmanabadi	39	C-0.024	3rd	*p* > 0.05	AA	
	India	Sangamneri	43	C-0.070	1st	*p* > 0.05	AA	
	India	Sangamneri	43	C-0.070	2nd	*p* > 0.05	AA	
	India	Sangamneri	43	C-0.070	3rd	*p* > 0.05	AA	
3	India	Black Bengal goats	158	C-0.035				[48]
	India	Barbari	50	C-0.040				
	India	Beetal	28	C-0.018				
	India	Ganjam	50	C-0.030				
	India	Jhakrana	14	C-0.036				
	India	Osmanabadi	41	C-0.024				
	India	Sangamneri	50	C-0.070				
4	China	Jining Grey goats	177	C-0.350	1st	*p < 0.5*	AC, CC	[50]
	China	Guizhou White goats	71	C-0.373				
	China	Boer goats	47	C-0.170				
	China	Liaoning cashmere goats	40	C-0.200				
5	India	Black Bengal goats	158	C-0.040		*p* > 0.05	AC	[23]
6	China	Lubei White goats	90	C-0.151		*p* > 0.05	CC	[58]
	China	Yimeng Black goats	80	C-0.182		*p* > 0.05	AA	
7	China	Yangtse River Delta White goats	105	C-0.262				[60]
	China	Huanghuai goats	40	C-0.150				
	China	Boer goats	35	C-0.086				
8	Iran	Markhoz goats	120	A-0.377		*p* < 0.05	AC	[61]
9	Iran	Markhoz goats	164	A-0.314		*p* > 0.05	AA	[46]

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
