# Peer review of "Genetic Effects of Single Nucleotide Polymorphisms in the Goat GDF9 Gene on Prolificacy: True or False Positive?"

_animals, 2019, doi:10.3390/ani9110886_

Round 1
Reviewer 1 Report
The scientific content of the article merits publication in the journal. However, there are major issues with the language of the paper. My serious advice to the authors is to seek help of English speaking experts to improve the language.
I am highlighting some of the sentences which need revision:
Lines 38-41: According to all related previous studies and the NCBI database, a total of 45 single nucleotide polymorphisms (SNPs) have been detected in the goat GDF9 gene, but which one or which ones have important effects on goat fecundity is still confused now. Lines 46-48: Despite there was a great many SNPs in the goat GDF9 gene, only 15 SNPs had been identified in more than 30 goat breeds worldwide and showed different effects on the litter size. Lines 49-52: Results showed that three non-synonymous SNPs A240V, Q320P, V397I, and three synonymous L61L, N121N, and L141L played a “true” role in the litter size trait in many goat breeds around the world, while the related pathways and mechanisms were need further research. Lines 58-61: Goat milk has the characteristics of richer casein micelles, less lactose and higher proportion of small milk fat globules, which make it has greater similarity to human milk and easier for human absorption and utilization. Lines 68-70: These phenomena can be seen that goat products play a key role in people’s lives, which means increasing goat stocks and improving the goat production performance are particularly important and meaningful. Lines 85-87: An 18-bp indel in the 5’-UTR of pig SRY-box transcription factor 9 (SOX9) gene had significantly association with male testis weight (P < 0.05) and control the expression of SOX9 Lines 124-127: However, studies on the mechanism of the goat GDF9 gene in regulating various normal physiological and metabolic processes of ovaries and how to maintain the normal function of ovaries are still scarce, and further studies are required to clarify these topics. Lines 131-135: It is intriguingly that each of these 15 mutations showed significance association with goat litter size trait, but in different researches the same SNP mutation exhibited inconsistent characteristics, such as different positive allele and mutation frequencies, especially diverse effects on lambing. Lines 165-168: Therefore, it is of great significance to explore the goat GDF9 gene as a high fecundity candidate gene and select some important SNP markers on it, which like using FecB as a key marker gene to improve the fecundity of sheep. Lines 174-178: These results make it confused to find a really and effective SNP locus using in goat molecule breeding. Therefore, based on the depth and width of previous studies, we selected these 15 controversial and potentially relevant valuable SNPs, and divided they into the following four parts to systematic and comprehensive summarize and discuss. Lines 185-187: Markhoz goats, a breed displaying variation in the litter size and an endangered status in Iran, were used to detect the mutation V397I in 164 individuals via the PCR-RFLP method. Lines 202-203: That’s might due to the small simple size and the lower for individual selection intensity in above seven goat breeds. Lines 221-222: This seems to be due to a small number of effective mutant populations. Lines 230-233: However, in Xinong Saanen dairy, Guanzhong dairy, Boer, Anhui White, Big foot black, and Jintang black goats, the individuals with “AA” genotype had better litter size (P < 0.05), which can be said this SNP showed positive effects on litter size. Lines 235-243: It is noticeable, however, recently Mahmoudi et al. (2019) used meta-analysis which conducted by pooling results from parts of above published researches to investigate effects of V397I on litter size using four different genetic models, and it found the effects were inconsistence by different model but the sensitivity analysis suggested this SNP was negative on goat litter size trait [60]. These results suggested that V397I seems playing a “true” role in goat reproductive traits, but the inconsistent studies results might due to this mutation is a minor quantitative trait nucleotide (QTN) which is linked with other stabilized QTNs during the long-term selection of evolution in goat breeds with different economic performances. Lines 247-250: Zhao et al. (2016) pointed out the genotypes of this SNP were significantly distributed in high- (more than two litters for a continuous six years) and low-fecundity Inner Mongolia cashmere goats via sequencing [51]. Lines 263-270: By contrast, Arefnejad et al. (2018) found that Q320P significantly negative affected the litter size with a reference allele “A” in 120 Markhoz goats [46]. However, Shokrollahi and Morammazi (2018) recently adopted a larger sample size of Markhoz goats (n = 146) for research and found that the frequence of mutant allele was similar (about 0.6) but it affected the litter size trait anymore [62], suggesting that the sample size might be a factor that has influenced accurate conclusions made by researchers and larger experimental populations are needed for further study to improve the reliability and accuracy of the results. Lines 278-280: Therefore, based on the above studies on goat breeds from China, Iran and India, it can be seen that the Q320P site showed an unstable effect on the goat litter size trait, which would change with the changes of goat breeds.
Same is the story for the entire paper.
Author Response
Dear editor and anonymous reviewers,
We greatly appreciate the anonymous reviewers for your careful review and constructive comments (Manuscript ID: animals-614458). We have read comments carefully and tried our best to revise the manuscript (animals-614458R1), and we hope that the revision meet your approval. According to the reviewer’s comments, we have corrected the language of the manuscript. The current revision has been re-edited and re-corrected by the professional editors at MDPI English editing (English editing ID: English-9627; see the certificate of English editing). Hence, we think it is better than the previous version.
Here, we have listed the point-by-point responses to your detailed comments and suggestions (with blue). As follows:
Responds to the reviewer’s comments:
Reviewer #1:
[Comment 1]
The scientific content of the article merits publication in the journal. However, there are major issues with the language of the paper. My serious advice to the authors is to seek help of English speaking experts to improve the language.
[Response 1]
Thank you for your suggestion.
According to your comment, we have corrected the language of the manuscript. In the current revision, many corrections of language, grammar and structure throughout the current revision, have been re-edited and re-corrected by the professional editors at MDPI English editing (English editing ID: English-9627;) (See the certificate of English editing in the file). This editing company offers professional English language editing, which editing by the native speakers and can to improve the expression at any stage prior to publication. Hence, we think that the quality of the text improves substantially.
[Comment 2]
Lines 38-41: According to all related previous studies and the NCBI database, a total of 45 single nucleotide polymorphisms (SNPs) have been detected in the goat GDF9 gene, but which one or which ones have important effects on goat fecundity is still confused now.
[Response 2]
Thanks.
According to your opinion and the result after language modification, we modified this sentence as follow: “According to the relevent previous studies and the NCBI database, a total of 45 single nucleotide polymorphisms (SNPs) have been detected in the goat GDF9 gene, but which one or which ones have important effects on goat fecundity is still uncertain”
[Comment 3]
Lines 46-48: Despite there was a great many SNPs in the goat GDF9 gene, only 15 SNPs had been identified in more than 30 goat breeds worldwide and showed different effects on the litter size.
[Response 3]
Thanks.
According to your opinion and the result after language modification, we modified this part as follow: “Despite there was a great many SNPs in the goat GDF9 gene, only 15 SNPs have been identified in more than 30 goat breeds worldwide and they showed different effects on the litter size. ”
[Comment 4]
Lines 49-52: Results showed that three non-synonymous SNPs A240V, Q320P, V397I, and three synonymous L61L, N121N, and L141L played a “true” role in the litter size trait in many goat breeds around the world, while the related pathways and mechanisms were need further research.
[Response 4]
Thank you for your suggestion.
According to your opinion and the result after language modification, we revised this part as follow: “Results showed that three non-synonymous SNPs A240V, Q320P and V397I and three synonymous ones L61L, N121N and L141L played a “true” role in the litter size trait in many goat breeds around the world, however, the regulatory mechanisms still need further research. ”
[Comment 5]
Lines 58-61: Goat milk has the characteristics of richer casein micelles, less lactose and higher proportion of small milk fat globules, which make it has greater similarity to human milk and easier for human absorption and utilization.
[Response 5]
Thank you for your advice.
Based on you suggestion and the result after language modification, we corrected this sentence into “Goat milk has the characteristics of richer casein micelles, less lactose and higher proportion of small milk fat globules, which makes it more similar to human milk and easier for human absorption and utilization.”
[Comment 6]
Lines 68-70: These phenomena can be seen that goat products play a key role in people’s lives, which means increasing goat stocks and improving the goat production performance are particularly important and meaningful.
[Response 6]
Thank you for you suggestion.
According to your suggestion, we corrected this sentence into“It can be seen that goat products play a key role in people’s lives from above phenomena, which means increasing goat stocks and improving the goat production performance are particularly important and meaningful. ”
[Comment 7]
Lines 85-87: An 18-bp indel in the 5’-UTR of pig SRY-box transcription factor 9 (SOX9) gene had significantly association with male testis weight (P < 0.05) and control the expression of SOX9
[Response 7]
Thanks.
Based on you suggestion and the result after language modification, we corrected this sentence into“ An 18-bp indel in the 5’-UTR of pig SRY-box transcription factor 9 (SOX9) gene was significantly associated with boar testis weight (P < 0.05) and controlled the expression of SOX9”
[Comment 8]
Lines 124-127: However, studies on the mechanism of the goat GDF9 gene in regulating various normal physiological and metabolic processes of ovaries and how to maintain the normal function of ovaries are still scarce, and further studies are required to clarify these topics.
[Response 8]
Thanks!
According to your opinion and the result after language modification, we revised this part as follow: “However, studies on the mechanism of the goat GDF9 gene in regulating various normal physiological and metabolic processes of ovaries and how to maintain the normal function of ovaries are still scarce, and further studies are required to clarify these issues”
[Comment 9]
Lines 131-135: It is intriguingly that each of these 15 mutations showed significance association with goat litter size trait, but in different researches the same SNP mutation exhibited inconsistent characteristics, such as different positive allele and mutation frequencies, especially diverse effects on lambing.
[Response 9]
Thank you for your useful advice.
Based on your suggestions, we revised this sentence as follow: “An intriguing funding is that each of these 15 SNPs showed significance association with goat litter size trait, but in different researches the same SNP mutation exhibited inconsistent characteristics, such as different positive allele and mutation frequencies, especially diverse effects on lambing.”
[Comment 10]
Lines 165-168: Therefore, it is of great significance to explore the goat GDF9 gene as a high fecundity candidate gene and select some important SNP markers on it, which like using FecB as a key marker gene to improve the fecundity of sheep.
[Response 10]
Thank you for your suggestion.
According to your suggestion and the result after language modification, we modified this part of content as: “Therefore, it is of great significance to explore and select some important SNP markers in the goat GDF9 gene, which like using Q249R mutation of FecB as a key marker gene to improve the fecundity of sheep. ”
[Comment 11]
Lines 174-178: These results make it confused to find a really and effective SNP locus using in goat molecule breeding. Therefore, based on the depth and width of previous studies, we selected these 15 controversial and potentially relevant valuable SNPs, and divided they into the following four parts to systematic and comprehensive summarize and discuss.
[Response 11]
Thanks.
According to your opinion, we corrected it into “These results make it confused to find a real and effective SNP locus for goat molecule breeding. Therefore, based on the depth and width of previous studies, we selected these 15 controversial and potentially relevant valuable SNPs, and divided them into the following four parts for systematic and comprehensive summary and discussion.”
[Comment 12]
Lines 185-187: Markhoz goats, a breed displaying variation in the litter size and an endangered status in Iran, were used to detect the mutation V397I in 164 individuals via the PCR-RFLP method.
[Response 12]
Thank you for your advice.
According to your opinion and the company's opinion of language modification, we have rewritten this sentence, and the modification result is as follows: “Markhoz goats, an endangered Iran breed with the characteristic of variation in the litter size, were used to detect the mutation V397I in 164 individuals via the PCR-RFLP method.”
[Comment 13]
Lines 202-203: That’s might due to the small simple size and the lower for individual selection intensity in above seven goat breeds.
[Response 13]
Thanks.
According to your opinion and the company's opinion of language modification, we revised this sentence as following : “That may be due to the small sample size and low for individual selection intensity in above seven goat breeds. ”
[Comment 14]
Lines 221-222: This seems to be due to a small number of effective mutant populations.
[Response 14]
Thanks.
According to your suggestion, we revised this sentence as following :“This presumably due to the number of effective mutant populations was small. ”
[Comment 15]
Lines 230-233: However, in Xinong Saanen dairy, Guanzhong dairy, Boer, Anhui White, Big foot black, and Jintang black goats, the individuals with “AA” genotype had better litter size (P < 0.05), which can be said this SNP showed positive effects on litter size.
[Response 15]
Thank you for your advice.
According to your opinion and the company's opinion of language modification, we have rewritten this sentence as follows: “However, in Xinong Saanen dairy, Guanzhong dairy, Boer, Anhui White, Big foot black, and Jintang black goats, the litter size of individuals with “AA” genotype was better than that of individuals with “GG” (P < 0.05), which could be concluded that V397I mutation showed positive effects on the litter size in these 6 goat breeds.”
[Comment 16]
Lines 235-243: It is noticeable, however, recently Mahmoudi et al. (2019) used meta-analysis which conducted by pooling results from parts of above published researches to investigate effects of V397I on litter size using four different genetic models, and it found the effects were inconsistence by different model but the sensitivity analysis suggested this SNP was negative on goat litter size trait [60]. These results suggested that V397I seems playing a “true” role in goat reproductive traits, but the inconsistent studies results might due to this mutation is a minor quantitative trait nucleotide (QTN) which is linked with other stabilized QTNs during the long-term selection of evolution in goat breeds with different economic performances.
[Response 16]
Thank you for your suggestions.
Based on you suggestion and the result after language modification, we corrected this sentence into “It is noticeable, however, that recently Mahmoudi et al. (2019) used meta-analysis to investigate effects of V397I on litter size, which used four different genetic models to calculate the pooling results from parts of above published researches, and it is found the effects were inconsistent in different model but the sensitivity analysis suggested that this SNP affected negatively on goat litter size trait [60]. These results suggested that V397I seems to play a “true” role in goat reproductive traits, but the inconsistent studies results might be due to the fact that this mutation is a minor quantitative trait nucleotide (QTN) which is linked with other stabilized QTNs during the long-term selection of evolution in goat breeds with different economic performances.”
[Comment 17]
Lines 247-250: Zhao et al. (2016) pointed out the genotypes of this SNP were significantly distributed in high- (more than two litters for a continuous six years) and low-fecundity Inner Mongolia cashmere goats via sequencing [51].
[Response 17]
Thank you for you valuable advice!
According to your opinion and the company's opinion of language modification, we have revised this sentence as follows: “Zhao et al. (2016) pointed out that the “AA” and “CC” genotypes were differentially distributed (P < 0.05) in high- (more than two litters for a continuous six years) and low-fecundity Inner Mongolia cashmere goats via sequencing [51]”
[Comment 18]
Lines 263-270: By contrast, Arefnejad et al. (2018) found that Q320P significantly negative affected the litter size with a reference allele “A” in 120 Markhoz goats [46]. However, Shokrollahi and Morammazi (2018) recently adopted a larger sample size of Markhoz goats (n = 146) for research and found that the frequence of mutant allele was similar (about 0.6) but it affected the litter size trait anymore [62], suggesting that the sample size might be a factor that has influenced accurate conclusions made by researchers and larger experimental populations are needed for further study to improve the reliability and accuracy of the results.
[Response 18]
Thank you for you valuable advice!
According to your opinion and the result after language modification, we revised this part as follow:“By contrast, Arefnejad et al. (2018) found that Q320P significantly affected the litter size in a negative way with a reference allele “A”in 120 Markhoz goats [46]. However, Shokrollahi and Morammazi (2018) recently adopted a larger sample size of Markhoz goats (n = 146) for research and found that the frequence of mutant allele was similar (about 0.6) but it didn’t affected the litter size trait anymore [62], suggesting that the sample size might be a factor that has influenced the conclusions made by the researchers and larger experimental populations are needed for further study to improve the reliability and accuracy of the results”
[Comment 19]
Lines 278-280: Therefore, based on the above studies on goat breeds from China, Iran and India, it can be seen that the Q320P site showed an unstable effect on the goat litter size trait, which would change with the changes of goat breeds.
[Response 19]
Thank you for you advice!
According to your opinion and the result after language modification, we corrected this part as follow:“Therefore, based on the results of studies in Chinese, Iranian and Indian goats, the effect of Q320P locus on litter size trait was unstable and varied with the change of goat breeds.”
Thanks very much for your comments and attention to our paper.
With best Regards!
Yours Sincerely,
M.D. X.Y. Wang,
Ph.D. X,Y. Lan and Ph.D. H, J. Zhu (corresponding author) .
College of Animal Science and Technology,
Northwest A&F University,
Yangling, Shaanxi 712100, China

Reviewer 2 Report
In general, the authors improved the manuscript significantly. However, it is important to add in addition to cover letter, point by point answer for each question raised by the reviewer and highlight in this letter what you actually changed or added. The authors addressed all my request, they mentioned Table 1, Table 2 and Table 3, but i could not find any table in the main manuscript. I would like to see the three Tables which you mentioned in the text. In line 222, you refer to Table 3, it is Table 1 or 3??? In line 322, please add space between analysis and result. In conclusion section, line 393 and 396, please do not refer to any table in the conclusion. Only conclude your main finding (delete as shown in table 1, 2, and S1) and table 1 and 2.Author Response
A cover letter with responses to the reviewers' comments on animals-614458R1
Dear editor and anonymous reviewers,
We greatly appreciate the anonymous reviewers for your careful review and constructive comments (Manuscript ID: animals-614458). We have read comments carefully and tried our best to revise the manuscript (animals-614458R1), and we hope that the revision meet your approval.
According to the reviewer’s comments, we carefully checked and corrected some errors and unreasonable expressions in the paper, and increased the accuracy and readability of the revised manuscript (animals-614458R1). Moreover, all tables were separately summarized into a word document and uploaded to the Animals submission system together with the manuscript document. Hence, we think it is better than the previous version.
Here, we have listed the point-by-point responses to your detailed comments and suggestions (with blue). As follows:
Responds to the reviewer’s comments:
Reviewer #2:
[Comment 1]
In general, the authors improved the manuscript significantly. However, it is important to add in addition to cover letter, point by point answer for each question raised by the reviewer and highlight in this letter what you actually changed or added.
[Response 1]
Thank you for your suggestion.
Since the editor of Animals magazine had not received the comments from reviewers last time, it directly decided to ask us to re-submit the paper after making major modifications. Therefore, I could not give a reply to the questions point-to-point raised by the reviewers in the last reversion. In this reversion (animals-614458R1), we provided this coverletter to answer the questions raised by you, and we hope that the revision meet your approval.
[Comment 2]
The authors addressed all my request, they mentioned Table 1, Table 2 and Table 3, but i could not find any table in the main manuscript. I would like to see the three Tables which you mentioned in the text.
[Response 2]
Thanks.
Due to the excessive number of columns in the table, horizontal typesetting was adopted for page layout, which was different from the vertical layout of the manuscript. Therefore, all the tables were separately arranged into a word document (named Tables) and submitted. The specific contents of all tables are as follows:
Table 1 All non-synonymous SNP loci (verified and predicted) within goat GDF9 gene
SNP locus |
Rs of the SNP |
Breed and simple size |
Mutant allele frequency |
Effect of the mutant allele on the litter size trait |
P values |
reference |
g.1972T>C/ c.149T>C/p.L50P |
- |
Lezhi black goats, n=6 |
- |
- |
- |
[70] |
|
- |
Tibetan goats, n=6 |
- |
- |
- |
|
g.3665C>T/ c.719C>T/ p.A240V |
rs637835524 |
Henan dairy goats,n=166 |
0.075 |
Postive |
P < 0.05 |
[53] |
|
|
Yaoshan goats, n=98 |
0.036 |
- |
- |
|
|
|
Taihang Black goats, n=102 |
0.024 |
- |
- |
|
g.3665C>T/ c.719C>T/ p.A240V |
rs637835524 |
Jining Grey goats,n=234 |
0.092 |
- |
P > 0.05 |
[58] |
|
|
Lubei White goats, n=90 |
0 |
- |
P > 0.05 |
|
|
|
Yimeng Black goats, n=80 |
0.116 |
- |
P > 0.05 |
|
g.3764C>T/ c.818C>T/ p.A273V |
rs662668357 |
Black Bengal goats, n=158 |
026 |
- |
P > 0.05 |
[23] |
g.3764C>T/ c.818C>T/ p.A273V |
rs662668357 |
Black Bengal goats, n=110 |
0.34 |
- |
P > 0.05 |
[47] |
|
|
Barbari, n=49 |
0.082 |
- |
P > 0.05 |
|
|
|
Beetal, n=28 |
0 |
- |
P > 0.05 |
|
|
|
Ganjam, n=50 |
0.04 |
- |
P > 0.05 |
|
|
|
Jhakrana, n=14 |
0 |
- |
P > 0.05 |
|
|
|
Osmanabadi, n=41 |
0 |
- |
P > 0.05 |
|
|
|
Sangamneri, n=50 |
0.04 |
- |
P > 0.05 |
|
g.3764C>T/ c.818C>T/ p.A273V |
rs662668357 |
Black Bengal goats, n=158 |
0.256 |
- |
- |
[48] |
|
|
Barbari, n=50 |
0.06 |
- |
- |
|
|
|
Beetal, n=28 |
0 |
- |
- |
|
|
|
Ganjam, n=50 |
0.02 |
- |
- |
|
|
|
Jhakrana, n=14 |
0 |
- |
- |
|
|
|
Osmanabadi, 41 |
0 |
- |
- |
|
|
|
Sangamneri, n=50 |
0.02 |
- |
- |
|
g.3905A>C/ c.959A>C/ p.Q320P |
rs645345606 |
Markhoz goats, n=120 |
0.623 |
Postive |
P < 0.05 |
[61] |
g.3905A>C/ c.959A>C/ p.Q320P |
rs645345606 |
Markhoz goats, n=164 |
0.686 |
- |
P > 0.05 |
[46] |
g.3905A>C/ c.959A>C/ p.Q320P |
rs645345606 |
Black Bengal goats, n=110 |
0.032 |
- |
P > 0.05 |
[47] |
|
|
Barbari, n=49 |
0.041 |
- |
P > 0.05 |
|
|
|
Beetal, n=27 |
0 |
- |
P > 0.05 |
|
|
|
Ganjam, n=50 |
0.04 |
- |
P > 0.05 |
|
|
|
Jhakrana, n=13 |
0 |
- |
P > 0.05 |
|
|
|
Osmanabadi, n=41 |
0.024 |
- |
P > 0.05 |
|
|
|
Sangamneri, n=50 |
0.07 |
- |
P > 0.05 |
|
g.3905A>C/ c.959A>C/ p.Q320P |
rs645345606 |
Black Bengal goats, n=158 |
0.035 |
- |
- |
[48] |
|
|
Barbari, n=50 |
0.040 |
- |
- |
|
|
|
Beetal, n=28 |
0.018 |
- |
- |
|
|
|
Ganjam, n=50 |
0.030 |
- |
- |
|
|
|
Jhakrana, n=14 |
0.036 |
- |
- |
|
|
|
Osmanabadi, n=41 |
0.024 |
- |
- |
|
|
|
Sangamneri, n=50 |
0.070 |
- |
- |
|
g.3905A>C/ c.959A>C/ p.Q320P |
rs645345606 |
Jining Grey goats, n=177 |
0.350 |
Postive |
P < 0.05 |
[50] |
|
|
Guizhou White goats, n=71 |
0.373 |
- |
P > 0.05 |
|
|
|
Boer goats, n=47 |
0.70 |
- |
P > 0.05 |
|
|
|
Liaoning cashmere goats, n=40 |
0.200 |
- |
P > 0.05 |
|
g.3905A>C/ c.959A>C/ p.Q320P |
rs645345606 |
Black Bengal goats, n=158 |
0.040 |
- |
P > 0.05 |
[23] |
g.3905A>C/ c.959A>C/ p.Q320P |
rs645345606 |
Lubei White goats, n=90 |
0151 |
- |
P > 0.05 |
[58] |
|
|
Yimeng Black goats, n=80 |
0.182 |
- |
P > 0.05 |
|
g.3905A>C/ c.959A>C/ p.Q320P |
rs645345606 |
Yangtse River Delta White goats, n=105 |
0.262 |
- |
- |
[60] |
|
|
Huanghuai goats, n=40 |
0.150 |
- |
- |
|
|
|
Boer goats, n=35 |
0.086 |
- |
- |
|
g.3905A>C/ c.959A>C/ p.Q320P |
rs645345606 |
Shaanbei white cashmere goats, n=1511 |
0.286 |
Postive |
P < 0.05 |
[45] |
g.4135A>G/c.1189A>G/ p.V397I |
rs637044681 |
Shaanbei white cashmere goats, n=1511 |
0.523 |
Negtive |
P < 0.05 |
[45] |
g.4135A>G/c.1189A>G/ p.V397I |
rs637044681 |
Markhoz goats, n=164 |
0.314 |
- |
P > 0.05 |
[46] |
g.4135A>G/c.1189A>G/ p.V397I |
rs637044681 |
Xinong Saanen dairy goats, n=241 |
0.290 |
Postive |
P < 0.05 |
[49] |
|
|
Guanzhong dairy goats, n=197 |
0290 |
Postive |
P < 0.05 |
|
|
|
Boer goats, n=203 |
0.290 |
Postive |
P < 0.05 |
|
g.4135A>G/c.1189A>G/ p.V397I |
rs637044681 |
Jining Grey goats, n=178 |
0.83 |
- |
P > 0.05 |
[50] |
|
|
Guizhou White goats, n=71 |
0.96 |
- |
- |
|
Continued Table 1 |
||||||
|
|
Boer goats, n=47 |
0.81 |
- |
- |
|
|
|
Liaoning cashmere goats, n=40 |
0.64 |
- |
- |
|
g.4135A>G/c.1189A>G/ p.V397I |
rs637044681 |
Henan dairy goats, n=168 |
0.83 |
Postive |
P < 0.05 |
[53] |
|
|
Yaoshan goats, n=98 |
0.92 |
- |
- |
|
|
|
Taihang Black goats, n=102 |
0.89 |
- |
- |
|
g.4135A>G/c.1189A>G/ p.V397I |
rs637044681 |
Black Bengal goats, n=110 |
0.92 |
- |
P > 0.05 |
[47] |
|
|
Barbari, n=49 |
0.90 |
- |
P > 0.05 |
|
|
|
Beetal, n=28 |
0.64 |
- |
P > 0.05 |
|
|
|
Ganjam, n=50 |
0.93 |
- |
P > 0.05 |
|
|
|
Jhakrana, n=13 |
0.50 |
- |
P > 0.05 |
|
|
|
Osmanabadi, n=41 |
0.99 |
- |
P > 0.05 |
|
|
|
Sangamneri, n=50 |
0.78 |
- |
P > 0.05 |
|
g.4135A>G/c.1189A>G/ p.V397I |
rs637044681 |
Black Bengal goats, n=158 |
0.89 |
- |
- |
[48] |
|
|
Barbari, n=50 |
0.90 |
- |
- |
|
|
|
Beetal, n=28 |
0.61 |
- |
- |
|
|
|
Ganjam, n=50 |
0.93 |
- |
- |
|
|
|
Jhakrana, n=14 |
0.54 |
- |
- |
|
|
|
Osmanabadi, n=41 |
0.84 |
- |
- |
|
|
|
Sangamneri, n=50 |
0.78 |
- |
- |
|
g.4135A>G/c.1189A>G/ p.V397I |
rs637044681 |
Inner Mongolia cashmere goats, n=761 |
0.68 |
Postive |
P < 0.05 |
[51] |
g.4135A>G/c.1189A>G/ p.V397I |
rs637044681 |
Big foot black goats, n=96 |
0.57 |
Negative |
P < 0.05 |
[52] |
|
|
Jintang black goats, n=81 |
0.32 |
Negative |
P < 0.05 |
|
g.4135A>G/c.1189A>G/ p.V397I |
rs637044681 |
Black Bengal goats, n=158 |
0.89 |
- |
P > 0.05 |
[23] |
g.4135A>G/c.1189A>G/ p.V397I |
rs637044681 |
Jining Grey goats, n=109 |
0.93 |
Postive |
P < 0.05 |
[59] |
|
|
Wendeng dairy goats, n=40 |
0.31 |
- |
- |
|
|
|
Liaoning cashmere goats, n=38 |
0.40 |
- |
- |
|
|
|
Beijing native goats, n=31 |
0.68 |
- |
- |
|
|
|
Boer goats, n=28 |
0.57 |
- |
- |
|
g.4135A>G/c.1189A>G/ p.V397I |
rs637044681 |
Anhui White goats, n=67 |
0.72 |
Negative |
P < 0.05 |
[54] |
g.4135A>G/c.1189A>G/ p.V397I |
rs637044681 |
Haimen goats, n=113 |
0.77 |
- |
P > 0.05 |
[55] |
|
|
Xuhuai goats, n=88 |
0.89 |
- |
P > 0.05 |
|
g.4135A>G/c.1189A>G/ p.V397I |
rs637044681 |
Laiwu Black goats, n=32 |
0.86 |
Postive |
P < 0.05 |
[56] |
|
|
Jining Grey goats, n=60 |
0.92 |
Postive |
P < 0.05 |
|
|
|
Laoshan dairy goats, n=55 |
0.40 |
- |
P > 0.05 |
|
g.4135A>G/c.1189A>G/ p.V397I |
rs637044681 |
Chongming White goats, n=75 |
0.80 |
- |
P >0.05 |
[57] |
g.4135A>G/c.1189A>G/ p.V397I |
rs637044681 |
Jining Grey goats, n=234 |
0.80 |
- |
P > 0.05 |
[58] |
|
|
Lubei White goats, n=90 |
0.68 |
Negative |
P < 0.05 |
|
|
|
Yimeng Black goats, n=80 |
0.85 |
Negative |
P < 0.05 |
|
g.1902C>G/c.79C>G/ p.P27R |
rs671913497 |
- |
- |
- |
- |
- |
g.2077C>G/c.254C>G/ p.A85G |
rs654628150 |
- |
- |
- |
- |
- |
g.3556G>C/ c.610G>C/ p.E204Q |
rs666975374 |
- |
- |
- |
- |
- |
Table 2 All synonymous SNP loci (verified and predicted) within goat GDF9 gene
SNP locus |
Rs of the SNP |
Breed and simple size |
Mutant allele frequency |
Effect of the mutant allele on the litter size trait |
P values |
reference |
g.1941C>T/ c.118C>T/p.G40G |
- |
Lezhi black goats, n=6 |
- |
- |
- |
[69] |
|
- |
Tibetan goats, n=6 |
- |
- |
- |
|
g.2006C>A/ c.183C>A/p.L61L |
rs669811820 |
Jining Grey goats, n=224 |
0.61 |
Negative |
P < 0.05 |
[63] |
|
|
Wendeng dairy goats, n=40 |
0.84 |
- |
- |
|
|
|
Liaoning cashmere goats, n=39 |
0.88 |
- |
- |
|
|
|
Beijing native goats, n=40 |
0.91 |
- |
- |
|
|
|
Boer goats, n=39 |
0.88 |
- |
- |
|
g.2006C>A/ c.183C>A/p.L61L |
rs669811820 |
Jining Grey goats, n=234 |
0.33 |
- |
P > 0.05 |
[58] |
|
|
Lubei White goats, n=90 |
0.20 |
- |
P > 0.05 |
|
|
|
Yimeng Black goats, n=80 |
0.35 |
Negative |
P < 0.05 |
|
g.2006C>A/ c.183C>A/p.L61L |
rs669811820 |
Anhui White goats, n=68 |
0.618 |
Negative |
P < 0.05 |
[54] |
g.2159C>T/c.336C>T/p.N112N |
- |
Jining Grey goats, n=224 |
0.39 |
Postive |
P < 0.05 |
[63] |
|
|
Wendeng dairy goats, n=40 |
0.16 |
- |
- |
|
|
|
Liaoning cashmere goats, n=39 |
0.12 |
- |
- |
|
|
|
Beijing native goats, n=40 |
0.09 |
- |
- |
|
|
|
Boer goats, n=39 |
0.12 |
- |
- |
|
g.2211G>A/c.387G>A/p.D129D |
- |
Assam hill goat, n=92 |
- |
- |
- |
[71] |
g.3441C>A/c.495C>A/p.S165S |
- |
Assam hill goat, n=92 |
- |
- |
- |
[71] |
g.3369G>A/ c.423G>A/p.L141L |
rs650650729 |
Jining Grey goats, n=178 |
0.104 |
- |
P > 0.05 |
[50] |
|
|
Guizhou White goats, n=71 |
0.021 |
- |
P > 0.05 |
|
|
|
Boer goats, n=47 |
0.192 |
- |
P > 0.05 |
|
|
|
Liaoning cashmere goats, n=40 |
0.363 |
- |
P > 0.05 |
|
g.3369G>A/ c.423G>A/p.L141L |
rs650650729 |
Jining Grey goats, n=109 |
0.087 |
Negative |
P < 0.05 |
[59] |
|
|
Wendeng dairy goats, n=40 |
0.313 |
- |
- |
|
|
|
Liaoning cashmere goats, n=38 |
0.605 |
- |
- |
|
|
|
Beijing native goats, n=31 |
0.097 |
- |
- |
|
|
|
Boer goats, n=28 |
0.304 |
- |
- |
|
g.3369G>A/ c.423G>A/p.L141L |
rs650650729 |
Laiwu Black goats, n=32 |
0.187 |
- |
P > 0.05 |
[56] |
|
|
Jining Grey goats, n=60 |
0.108 |
Negative |
P < 0.05 |
|
|
|
Laoshan dairy goats, n=55 |
0.709 |
- |
P > 0.05 |
|
g.3597G>T/ c.651G>T/p.T217T |
rs651511232 |
- |
- |
- |
- |
- |
Table 3 V397I mutation within GDF9 gene associated with litter size in global goat breeds.
Number |
Test method |
Country |
Breeds |
Sample size |
Minor allele frequency |
Parity |
P values |
Dominant genotype |
Reference |
1 |
PCR-RFLP & Sequencing |
China |
Shaanbei white cashmere goats |
1511 |
A-0.477 |
1st |
P<0.05 |
AG |
[45] |
2 |
PCR-RFLP & Sequencing |
China |
Xinong Saanen dairy goats |
241 |
G-0.290 |
1st |
P<0.05 |
AA,GA |
[49] |
|
PCR-RFLP & Sequencing |
China |
Xinong Saanen dairy goats |
241 |
G-0.290 |
2nd |
P<0.05 |
AA |
|
|
PCR-RFLP & Sequencing |
China |
Xinong Saanen dairy goats |
241 |
G-0.290 |
3rd |
P<0.05 |
AA,GA |
|
|
PCR-RFLP & Sequencing |
China |
Xinong Saanen dairy goats |
241 |
G-0.290 |
4th |
P>0.05 |
AA |
|
|
PCR-RFLP & Sequencing |
China |
Xinong Saanen dairy goats |
241 |
G-0.290 |
Average |
P<0.05 |
AA,GA |
|
|
PCR-RFLP & Sequencing |
China |
Guanzhong dairy goats |
197 |
G-0.290 |
1st |
P<0.05 |
|
|
|
PCR-RFLP & Sequencing |
China |
Guanzhong dairy goats |
197 |
G-0.290 |
2nd |
P<0.05 |
AA |
|
|
PCR-RFLP & Sequencing |
China |
Guanzhong dairy goats |
197 |
G-0.290 |
3rd |
P<0.05 |
AA,GA |
|
|
PCR-RFLP & Sequencing |
China |
Guanzhong dairy goats |
197 |
G-0.290 |
4th |
P>0.05 |
AA |
|
|
PCR-RFLP & Sequencing |
China |
Guanzhong dairy goats |
197 |
G-0.290 |
Average |
P<0.05 |
AA,GA |
|
|
PCR-RFLP & Sequencing |
China |
Boer goats |
203 |
G-0.290 |
1st |
P<0.05 |
|
|
|
PCR-RFLP & Sequencing |
China |
Boer goats |
203 |
G-0.290 |
2nd |
P<0.05 |
AA |
|
|
PCR-RFLP & Sequencing |
China |
Boer goats |
203 |
G-0.290 |
3rd |
P<0.05 |
AA,GA |
|
|
PCR-RFLP & Sequencing |
China |
Boer goats |
203 |
G-0.290 |
4th |
P>0.05 |
AA |
|
|
PCR-RFLP & Sequencing |
China |
Boer goats |
203 |
G-0.290 |
Average |
P<0.05 |
AA,GA |
|
3 |
PCR-RFLP & Sequencing |
China |
Jining Grey goats |
178 |
A-0.171 |
1st |
P>0.05 |
GG |
[50] |
|
PCR-RFLP & Sequencing |
China |
Guizhou White goats |
71 |
A-0.032 |
|
|
|
|
|
PCR-RFLP & Sequencing |
China |
Boer goats |
41 |
A-0.192 |
|
|
|
|
|
PCR-RFLP & Sequencing |
China |
Liaoning cashmere goats |
40 |
A-0.363 |
|
|
|
|
4 |
PCR-RFLP & Sequencing |
China |
Henan dairy goats |
168 |
A-0.173 |
1st |
P<0.05 |
GA |
[53] |
|
PCR-RFLP & Sequencing |
China |
Yaoshan goats |
98 |
A-0.082 |
|
|
|
|
|
PCR-RFLP & Sequencing |
China |
Taihang Black goats |
102 |
A-0.113 |
|
|
|
|
5 |
PCR-RFLP |
India |
Black Bengal goats |
110 |
A-0.180 |
1st |
P>0.05 |
GA |
[47] |
|
PCR-RFLP |
India |
Black Bengal goats |
110 |
A-0.180 |
2nd |
P>0.05 |
AA |
|
|
PCR-RFLP |
India |
Black Bengal goats |
110 |
A-0.180 |
3rd |
P>0.05 |
AA |
|
|
PCR-RFLP |
India |
Barbari |
49 |
A-0.102 |
1st |
P>0.05 |
GA |
|
|
PCR-RFLP |
India |
Barbari |
49 |
A-0.102 |
2nd |
P>0.05 |
GA |
|
|
PCR-RFLP |
India |
Barbari |
49 |
A-0.102 |
3rd |
P>0.05 |
GA |
|
|
PCR-RFLP |
India |
Beetal |
28 |
A-0.357 |
1st |
P>0.05 |
GG |
|
|
PCR-RFLP |
India |
Beetal |
28 |
A-0.357 |
2nd |
P>0.05 |
GG |
|
|
PCR-RFLP |
India |
Beetal |
28 |
A-0.357 |
3rd |
P>0.05 |
GG |
|
|
PCR-RFLP |
India |
Ganjam |
50 |
A-0.070 |
1st |
P>0.05 |
GG |
|
|
PCR-RFLP |
India |
Ganjam |
50 |
A-0.070 |
2nd |
P>0.05 |
GG |
|
|
PCR-RFLP |
India |
Ganjam |
50 |
A-0.070 |
3rd |
P>0.05 |
GG |
|
|
PCR-RFLP |
India |
Jhakrana |
13 |
A-0.500 |
1st |
P>0.05 |
|
|
|
PCR-RFLP |
India |
Jhakrana |
13 |
A-0.500 |
2nd |
P>0.05 |
|
|
|
PCR-RFLP |
India |
Jhakrana |
13 |
A-0.500 |
3rd |
P>0.05 |
|
|
|
PCR-RFLP |
India |
Osmanabadi |
41 |
A-0.012 |
1st |
P>0.05 |
GA |
|
|
PCR-RFLP |
India |
Osmanabadi |
41 |
A-0.012 |
2nd |
P>0.05 |
GG |
|
|
PCR-RFLP |
India |
Osmanabadi |
41 |
A-0.012 |
3rd |
P>0.05 |
GA |
|
|
PCR-RFLP |
India |
Sangamneri |
50 |
A-0.220 |
1st |
P>0.05 |
GA |
|
|
PCR-RFLP |
India |
Sangamneri |
50 |
A-0.220 |
2nd |
P>0.05 |
GA |
|
|
PCR-RFLP |
India |
Sangamneri |
50 |
A-0.220 |
3rd |
P>0.05 |
GG |
|
6 |
PCR-RFLP |
India |
Black Bengal goats |
158 |
A-0.108 |
|
|
|
[48] |
|
PCR-RFLP |
India |
Barbari |
50 |
A-0.100 |
|
|
|
|
|
PCR-RFLP |
India |
Beetal |
28 |
A-0.393 |
|
|
|
|
|
PCR-RFLP |
India |
Ganjam |
50 |
A-0.070 |
|
|
|
|
|
PCR-RFLP |
India |
Jhakrana |
14 |
A-0.464 |
|
|
|
|
|
PCR-RFLP |
India |
Osmanabadi |
41 |
A-0.159 |
|
|
|
|
|
PCR-RFLP |
India |
Sangamneri |
50 |
A-0.220 |
|
|
|
|
7 |
PCR-RFLP & Sequencing |
China |
Inner Mongolia cashmere goats |
761 |
A-0.320 |
1st |
P<0.05 |
GG |
[51] |
|
PCR-RFLP & Sequencing |
China |
Inner Mongolia cashmere goats |
761 |
A-0.320 |
2nd |
P<0.05 |
GG |
|
|
PCR-RFLP & Sequencing |
China |
Inner Mongolia cashmere goats |
761 |
A-0.320 |
3rd |
P>0.05 |
GG |
|
|
PCR-RFLP & Sequencing |
China |
Inner Mongolia cashmere goats |
761 |
A-0.320 |
4th |
P<0.05 |
GG |
|
|
PCR-RFLP & Sequencing |
China |
Inner Mongolia cashmere goats |
761 |
A-0.320 |
5th |
P<0.05 |
GG |
|
|
PCR-RFLP & Sequencing |
China |
Inner Mongolia cashmere goats |
761 |
A-0.320 |
6th |
P<0.05 |
GG |
|
|
PCR-RFLP & Sequencing |
China |
Inner Mongolia cashmere goats |
761 |
A-0.320 |
Average |
P<0.05 |
GG |
|
8 |
PCR-SSCP |
China |
Big foot black goats |
96 |
A-0.430 |
|
P<0.05 |
AA,GA |
[52] |
|
PCR-SSCP |
China |
Jintang black goats |
81 |
G-0.320 |
|
P<0.05 |
AA,GA |
|
9 |
PCR-RFLP |
India |
Black Bengal goats |
158 |
A-0.110 |
|
P>0.05 |
GA |
[23] |
10 |
PCR-SSCP & Sequencing |
China |
Jining Grey goats |
109 |
A-0.073 |
1st |
P<0.05 |
GG |
[59] |
|
PCR-SSCP & Sequencing |
China |
Wendeng dairy goats |
40 |
G-0.313 |
|
|
|
|
|
PCR-SSCP & Sequencing |
China |
Liaoning cashmere goats |
38 |
G-0.395 |
|
|
|
|
|
PCR-SSCP & Sequencing |
China |
Beijing native goats |
31 |
A-0.323 |
|
|
|
|
|
PCR-SSCP & Sequencing |
China |
Boer goats |
28 |
A-0.429 |
|
|
|
|
11 |
PCR-SSCP & Sequencing |
China |
Anhui White goats |
67 |
A-0.284 |
1st |
P<0.05 |
AA |
[54] |
12 |
PCR-SSCP & Sequencing |
China |
Haimen goats |
113 |
A-0.230 |
1st |
P>0.05 |
GA |
[55] |
|
PCR-SSCP & Sequencing |
China |
Xuhuai goats |
88 |
A-0.110 |
1st |
P>0.05 |
|
|
13 |
PCR-SSCP & Sequencing |
China |
Laiwu Black goats |
32 |
A-0.141 |
1st |
P<0.05 |
GG |
[56] |
|
PCR-SSCP & Sequencing |
China |
Jining Grey goats |
60 |
A-0.083 |
1st |
P<0.05 |
GG |
|
|
PCR-SSCP & Sequencing |
China |
Laoshan dairy goats |
55 |
A-0.600 |
1st |
P>0.05 |
GA |
|
14 |
PCR-SSCP & Sequencing |
China |
Chongming goats |
75 |
A-0.200 |
1st |
P>0.05 |
GA |
[57] |
15 |
PCR-SSCP & Sequencing |
China |
Lubei White goats |
90 |
A-0.325 |
1st |
P<0.05 |
AG,GG |
[58] |
|
PCR-SSCP & Sequencing |
China |
Yimeng Black goats |
80 |
A-0.148 |
1st |
P<0.05 |
AA |
|
16 |
PCR-RFLP |
Iran |
Markhoz goats |
164 |
G-0.314 |
1st |
P>0.05 |
|
[46] |
Table 4 Q320P mutation within GDF9 gene associated with litter size in global goat breeds.
Number |
Country |
Breeds |
Sample size |
Minor allele frequency |
Parity |
P values |
Dominant genotype |
Reference |
1 |
China |
Shaanbei white cashmere goats |
1511 |
C-0.286 |
1st |
P<0.05 |
AC,CC |
[45] |
2 |
India |
Black Bengal goats |
110 |
C-0.032 |
1st |
P>0.05 |
AC |
[47] |
|
India |
Black Bengal goats |
110 |
C-0.032 |
2nd |
P>0.05 |
AA |
|
|
India |
Black Bengal goats |
110 |
C-0.032 |
3rd |
P>0.05 |
AC |
|
|
India |
Barbari |
49 |
C-0.041 |
1st |
P>0.05 |
AA |
|
|
India |
Barbari |
49 |
C-0.041 |
2nd |
P>0.05 |
AA |
|
|
India |
Barbari |
49 |
C-0.041 |
3rd |
P>0.05 |
AA |
|
|
India |
Beetal |
27 |
C-0 |
1st |
P>0.05 |
AA |
|
|
India |
Beetal |
27 |
C-0 |
2nd |
P>0.05 |
AA |
|
|
India |
Beetal |
27 |
C-0 |
3rd |
P>0.05 |
AA |
|
|
India |
Ganjam |
46 |
C-0.040 |
1st |
P>0.05 |
AA |
|
|
India |
Ganjam |
46 |
C-0.040 |
2nd |
P>0.05 |
AA |
|
|
India |
Ganjam |
46 |
C-0.040 |
3rd |
P>0.05 |
AA |
|
|
India |
Jhakrana |
13 |
C-0 |
1st |
P>0.05 |
|
|
|
India |
Jhakrana |
13 |
C-0 |
2nd |
P>0.05 |
|
|
|
India |
Jhakrana |
13 |
C-0 |
3rd |
P>0.05 |
|
|
|
India |
Osmanabadi |
39 |
C-0.024 |
1st |
P>0.05 |
AA |
|
|
India |
Osmanabadi |
39 |
C-0.024 |
2nd |
P>0.05 |
AA |
|
|
India |
Osmanabadi |
39 |
C-0.024 |
3rd |
P>0.05 |
AA |
|
|
India |
Sangamneri |
43 |
C-0.070 |
1st |
P>0.05 |
AA |
|
|
India |
Sangamneri |
43 |
C-0.070 |
2nd |
P>0.05 |
AA |
|
|
India |
Sangamneri |
43 |
C-0.070 |
3rd |
P>0.05 |
AA |
|
3 |
India |
Black Bengal goats |
158 |
C-0.035 |
|
|
|
[48] |
|
India |
Barbari |
50 |
C-0.040 |
|
|
|
|
|
India |
Beetal |
28 |
C-0.018 |
|
|
|
|
|
India |
Ganjam |
50 |
C-0.030 |
|
|
|
|
|
India |
Jhakrana |
14 |
C-0.036 |
|
|
|
|
|
India |
Osmanabadi |
41 |
C-0.024 |
|
|
|
|
|
India |
Sangamneri |
50 |
C-0.070 |
|
|
|
|
4 |
China |
Jining Grey goats |
177 |
C-0.350 |
1st |
P<0.5 |
AC,CC |
[50] |
|
China |
Guizhou White goats |
71 |
C-0.373 |
|
|
|
|
|
China |
Boer goats |
47 |
C-0.170 |
|
|
|
|
|
China |
Liaoning cashmere goats |
40 |
C-0.200 |
|
|
|
|
5 |
India |
Black Bengal goats |
158 |
C-0.040 |
|
P>0.05 |
AC |
[23] |
6 |
China |
Lubei White goats |
90 |
C-0.151 |
|
P>0.05 |
CC |
[58] |
|
China |
Yimeng Black goats |
80 |
C-0.182 |
|
P>0.05 |
AA |
|
7 |
China |
Yangtse River Delta White goats |
105 |
C-0.262 |
|
|
|
[60] |
|
China |
Huanghuai goats |
40 |
C-0.150 |
|
|
|
|
|
China |
Boer goats |
35 |
C-0.086 |
|
|
|
|
8 |
Iran |
Markhoz goats |
120 |
A-0.377 |
|
P<0.05 |
AC |
[61] |
9 |
Iran |
Markhoz goats |
164 |
A-0.314 |
|
P>0.05 |
AA |
[46] |
[Comment 3]
In line 222, you refer to Table 3, it is Table 1 or 3???
[Response 3]
Thank you for your suggestion.
This refers to Table 3. Table1 mainly collated all the non-synonymous mutations of the goat GDF9 gene, which gave the information of the SNPs, studied breeds, simple size and the effect of the mutant allele on the litter size trait (Postive/Negative). Table3 mainly collated the results of V397I mutation in different studies, which was a refinement of Table1, from which we can intuitively see the dominant genotypes related to high lambing. Therefore, Table 3 is referred to here.
[Comment 4]
In line 322, please add space between analysis and result.
[Response 4]
Thanks.
Based on you suggestion, we added space between analysis and result.
[Comment 5]
In conclusion section, line 393 and 396, please do not refer to any table in the conclusion. Only conclude your main finding (delete as shown in table 1, 2, and S1) and table 1 and 2.
[Response 5]
Thank you for your advice.
Based on you suggestion, we have deleted “ as shown in table 1, 2, and S1” in the conclusion.
Thanks very much for your comments and attention to our paper.
With best Regards!
Yours Sincerely,
M.D. Xinyu Wang (wangxinyu6157@163.com),
Ph.D. Xianyong Lan (lanxianyong79@126.com) and Ph.D. Haijing Zhu (haijingzhu@yulinu.edu.cn), (corresponding author).
College of Animal Science and Technology,
Northwest A&F University,
Yangling, Shaanxi 712100, China.

This manuscript is a resubmission of an earlier submission. The following is a list of the peer review reports and author responses from that submission.
Round 1
Reviewer 1 Report
Dear authors
The manuscript has been improved by adding some references and content. The structure also follows a more logical pattern.
The content is however still not on the required standard and many statements are very vague. The aim of the manuscript as a whole is not clear, and I do not believe that it contributes significantly to current knowledge.
There are still numerous grammar and spelling errors.
Author Response
Dear editor and anonymous reviewers,
We greatly appreciate the anonymous reviewers for your careful review and constructive comments (Manuscript ID: animals-518329). We have read comments carefully and tried our best to revise the manuscript (animals-518329R1), and we hope that the revision meet your approval.
According to the reviewer’s comments, we have rechecked and rewrote some sentences in this manuscript to make it easy and clear to understand. Hence, we think it is better than the previous version.
Here, we have listed the point-by-point responses to your detailed comments and suggestions (with blue). As follows:
Responds to the reviewer’s comments:
Reviewer #1:
[Comment 1]
The manuscript has been improved by adding some references and content. The structure also follows a more logical pattern.
The content is however still not on the required standard and many statements are very vague. The aim of the manuscript as a whole is not clear, and I do not believe that it contributes significantly to current knowledge.
[Response 1]
Thank you for you comments.
Goat products are very popular at present, so it is very meaningful to effectively improve the production and reproduction ability of goats and provide more and better goat products. From the aspect of animal breeding, the method of marker-assisted selection has developed rapidly in recent years. This method can achieve early accurate selection of excellent breeding stock via selecting the polymorphisms mutations on candidate genes with high fertility as molecular markers. Therefore, it is very important to find the real effective molecular genetic mutation.
The GDF9 gene considered in this paper is one of the candidate genes of high fertility in livestock, and the genetic polymorphism of this gene in goats is relatively abundant at present. However, it is worth noting that the same mutation within goat GDF9 gene showed inconsistent characteristics among different goat breeds, especially the effect on litter size trait. Hence, the aim of this study was to summarize and sort the polymorphisms of GDF9 gene, mainly SNP loci, so as to find potential and effective polymorphic mutations.
As a result, 6 SNP loci (L61L, N121N, L141L, A240V, Q320P and V397I) were found to be closely related to the goat litter size trait from the previous studies. Among them, Q320P and V397I loci were also reported in our previous study, showing a strong linkage in cashmere goats. However, whether there are other linkage relationships among these loci is still unknown, and the exact molecular mechanisms by which the mutations work are also unknown. That’s might be why you think many statements are very vague in this manuscript. By comparing and collating the results of previous studies, we concluded that further studies on multiple goat breeds and large samples are needed to find out the real and effective molecular genetic mutations within goat GDF9 gene, and finally apply them to molecular breeding of goats. Therefore, the conclusion and summary of this paper can provide a potential direction and train of thought for subsequent research on the genetic selection of goat reproductive traits, which have a certain significance.
[Comment 2]
There are still numerous grammar and spelling errors.
[Response 2]
Thanks.
Based on the corrected version by MDPI English editing (English editing ID: English-9627; See the certificate of English editing), we checked and revised the grammar and spelling in this manuscript again. Hence, we think it is better than the previous version. Please refer to the highlight part in the revised manusucript.

Reviewer 2 Report
1. Line 20: and the NCBI database, and focused on analysis and discussion the relationship……..Add of before discussion.
2. Line 24: breeds and large simple size are required to study their ture effects on prolificacy……..ture?? Correct the spelling.
3. Line 31: The growth differentiation factor 9 (GDF9) gene a member of the……..Add is after gene.
4. Line 38: SNPs have been identifited in more than 30 goat breeds around the world…….. identifited? Correct the spelling.
5. Line 136: exhibit in the ovary……Replace exhibit with observed.
6. Line 138: Furthermore, GDF9 and FSH could maintained……..Replace maintained with maintain.
7. Line 146: Up until now……Delete up.
8. Line 154: that does not damage the protein structure via biological software prediction…….Delete via biological software prediction.
9. Line 155: Currently, it has been verified in a total of 32 goat breeds from across……Delete from.
10. Line 158: litter size and an endangered status in Iran, via the PCR-RFLP method……????
11. Line 231: breeds, which might due to the low mutation frequency (no greater than 5%)……Add be after might.
12. Line 278-278: However, this mutation has not been reported in any other goat breeds around the world so far, so there is a need to further……….breeds should be breed.
13. Line 309: In addition to the above reported controversial SNP mutation sites……..controversial does not seem to be an appropriate word.
14. Line 319: and A273V are three loci with that have received less attention and study…….delete with.
15. Rephrase these sentences:
· What is intriguing is that the results on the same mutation obtained from different researches were not in consistent, such as in the main allele, mutation frequencies and the effects on lambing, which were confusing and unable to find effective, applicable and valuable SNP mutations……Please rephrase these lines.
· The samples were fed in the Sanandaj Markhoz goat Performance Testing Station and they had not been selected based on lambing or other fertility traits in previous years…….Rephrase.
· Furthermore, association analysis indicated that the effect of parity on litter size was significant at V397I in the prolific Black Bengal, but the genotypes had no effects on litter size, which due to the small simple size and the lower for individual selection intensity.
16. All gene names should be in italics.
Author Response
Dear editor and anonymous reviewers,
We greatly appreciate the anonymous reviewers for your careful review and constructive comments (Manuscript ID: animals-518329). We have read comments carefully and tried our best to revise the manuscript (animals-518329R1), and we hope that the revision meet your approval.
According to the reviewer’s comments, we have rechecked and rewrote some sentences in this manuscript to make it easy and clear to understand. Hence, we think it is better than the previous version.
Here, we have listed the point-by-point responses to your detailed comments and suggestions (with blue). As follows:
Responds to the reviewer’s comments:
Reviewer #2:
[Comment 1]
Line 20: and the NCBI database, and focused on analysis and discussion the relationship……..Add of before discussion.
[Response 1]
Thanks.
According to your suggestion, we added “of” before “discussion”.
[Comment 2]
Line 24: breeds and large simple size are required to study their ture effects on prolificacy……..ture?? Correct the spelling.
[Response 2]
Thanks.
We revised the spelling errors. Now it corrected to “However, more goat breeds and large simple size are required to study their true effects on prolificacy.”
[Comment 3]
Line 31: The growth differentiation factor 9 (GDF9) gene a member of the……..Add is after gene.
[Response 3]
Thanks.
According to your suggestion, we added “is” after “gene”.
[Comment 4]
Line 38: SNPs have been identifited in more than 30 goat breeds around the world…….. identifited? Correct the spelling.
[Response 4]
Thank you for your suggestion.
We corrected “identifited” with “identified”.
[Comment 5]
Line 136: exhibit in the ovary……Replace exhibit with observed.
[Response 5]
Thank you for your suggestion.
According to your suggestion, we replaced “exhibit” with “observed” in this sentence.
[Comment 6]
Line 138: Furthermore, GDF9 and FSH could maintained……..Replace maintained with maintain.
[Response 6]
Thank you for your advice.
We replaced “maintained” with “maintain”.
[Comment 7]
Line 146: Up until now……Delete up.
[Response 7]
Thanks!
According to your suggestion, we have deleted “Up”.
[Comment 8]
Line154: that does not damage the protein structure via biological software prediction…….Delete via biological software prediction.
[Response 8]
Thanks.
we have deleted “via biological software prediction” in this sentence.
[Comment 9]
Line 155: Currently, it has been verified in a total of 32 goat breeds from across……Delete from.
[Response 9]
Thank you for your suggestion.
we have deleted “from” in this sentence.
[Comment 10]
Line 158: litter size and an endangered status in Iran, via the PCR-RFLP method……????
[Response 10]
Thank you for your suggestion.
According to your opinion, we rewritten this sentence as follow: Markhoz goats, a breed displaying variation in the litter size and an endangered status in Iran, were used to detected the mutation V397I in 164 individuals via the PCR-RFLP method.
[Comment 11]
Line 231: breeds, which might due to the low mutation frequency (no greater than 5%)……Add be after might.
[Response 11]
Thank you for your advice.
According to your opinion, we have added “be” after “might”.
[Comment 12]
Line 278-278: However, this mutation has not been reported in any other goat breeds around the world so far, so there is a need to further……….breeds should be breed.
[Response 12]
Thanks.
We replaced “breeds” with “breed”.
[Comment 13]
Line309: In addition to the above reported controversial SNP mutation sites……..controversial does not seem to be an appropriate word.
[Response 13]
Thanks.
According to your suggestion, we deleted “controversial ” in this sentence.
[Comment 14]
Line 319: and A273V are three loci with that have received less attention and study…….delete with.
[Response 14]
Thank you for your advice.
According to your suggestion, we deleted “with” in this sentence.
[Comment 15]
Rephrase these sentences:
· What is intriguing is that the results on the same mutation obtained from different researches were not in consistent, such as in the main allele, mutation frequencies and the effects on lambing, which were confusing and unable to find effective, applicable and valuable SNP mutations……Please rephrase these lines.
· The samples were fed in the Sanandaj Markhoz goat Performance Testing Station and they had not been selected based on lambing or other fertility traits in previous years…….Rephrase.
·Furthermore, association analysis indicated that the effect of parity on litter size was significant at V397I in the prolific Black Bengal, but the genotypes had no effects on litter size, which due to the small simple size and the lower for individual selection intensity.
[Response 15]
Thank you for your suggestions.
According to your suggestions, we rewritten above three sentences and the revised sentences are as following:
· It is intriguingly that each of these 15 mutations in different researches exhibited inconsistent characteristics, such as different main allele and mutation frequencies, expecially diverse effects on lambing. This make it more difficult for the goat breeders to find effective, applicable and valuable SNP mutations.
· These samples were fed in Sanandaj Markhoz goat Performance Testing Station, located in the Kurdistan province of Iran, and no selection on reproduction traits was performed over previous years.
· Furthermore, none of these three studies found the mutation V397I had significant effects on litter size, while in the prolific Black Bengal, Ahlawat et al. found the effect of parity on litter size trait was significant for V397I mutation [42]. That’s might due to the small simple size and the lower for individual selection intensity in above seven goat breeds.
[Comment 16]
All gene names should be in italics.
[Response 16]
Thanks!
We have recheck all gene names in the manuscript and now we make sure they are all in italics.

Reviewer 3 Report
The authors review the relationships between the polymorphisms within the GDF9 gene and reproductive traits using the 45 SNPs from the previous studies an NCBI database. The idea in general of reviewing this gene and their importance in goat breeds is good. However, the authors are not able to present it in good manner and provide us nice story out of that.
The manuscript is not clear and are not well written and it needs major revision before could be considered it in publication.
Below I summarized general points that should be considered at least to improve the manuscript:
The abstract is not well written and it contains more introduction stuff. The abstract should be revised and write the main objectives of this study, material and method used, most important results and conclusion. You can write in clear way why only you choose some of this SNPs and focus your review and discussion on it? The sub heading will be focus first on Non-synonymous SNPs, then synonymous SNPs and so on…. For each SNP you consider please give the locations in the new goat reference genome, then the scientists will be benefit out of that instead of giving different name of each SNP. It is important to make different table to present your review, instead to make big excel table in Supplementary material. For example Table 1 for Non-synonymous SNPs, Table 2 for synonymous SNPs and Table 3 for other SNPs in the regulatory region and intronic region. These tables which I mentioned in the previous point make your work clear to reader. It should contains the Name of SNP, position in the new reference goat genome, the rs of the SNP, reference and mutant allele, Allele frequency of the mutant allele, the effect of the mutant allele if they do association analysis, in which breed, in which study. Try to be homogenized when you write the story of each SNP, for e.g, location of the SNP in which exon, focus on the frequency of the mutant allele and in which breed even you talked about the range, and the effect of the mutant allele in the litter size. It is positive effect or negative effect, what is the magnitude of the effect? What is the significant p-value? Try to be discuss more the significant effects and make it clear at least to get the point directly!!! You are not even consider the effect of the genotypes, if it is homo or hetero and so on!!!! In the discussion part if you can discuss also if there is similar effect may be in sheep or cattle. In the title you write true or false positive, however you are not conclude it in your conclusion. If the effect of these SNPs on prolificacy is true or false? In line 12, page 1; please add the full address for the corresponding authors, not only the E-mail address. In line 22, page 1; you can write the non-Synonymous SNPs first and then Synonymous SNPs and consider the order like that in the whole manuscript. In line 42 in the keywords, only mentioned SNP and delete polymorphisms. In the introduction part line 47, make it to sentences, you considered goat milk with their important nutritional value because it has less milk fat, and the protein digestion. Then you talk about importance of goat milk in dairy products like cheese and yoghurt…etc. In the introduction part line 58, please add references after quality. In the introduction part line 70, no space between analysis and identified. In the introduction part line 75, when you mentioned the name of the gene in the first time, please write the full name and the abbreviation between brackets, then you can used the abbreviation after that in your text. In the introduction part line 80, please specify the breed not talk in general. And the same situation in line 82, which goat breed, please be specific. In general, when you mentioned the name of the reference in the text please write the year also after et al. In line 146, you talked about the different physiological effects which is confusing you said. Which type of confusing, make everything in clear way. In line 155, were used to detect instead of to detected. From line 160 to 165, you mentioned three different number after n, I could not follow you at all. Please in general try to talk and concentrate in the most significant effects of these SNPs on the reproductive traits, which type of effect? And the magnitude of the effect!!! Please try to be specific either used mutant allele or reference allele, sometimes you used major allele (both reference and mutant could be major allele or minor allele in the population, that make me sometimes lost to follow you. In line 230, please let me know how you draw this conclusion? It has no effect or????!!!!! In line 236, of the mutant allele instead of mutation. In line 242, which type of effect this SNP has in the first-born litter size? In line 256, again as I told you before either focus on mutant or reference allele in your manuscript? Which one has a positive effect on the reproductive performance of the goat? In line 304, the other mutation in GDF9 gene has a real and significant effect on reproductive traits? If yes in which direction? You talked always about the small sample size if there is no effect for the SNP? Did you recommended special sample size to find association analysis even in different goat populations??? What is your recommendation in general and your conclusion idea after you reviewing all these papers regarding the effect of this gene? What is the next step after to take benefit out of your manuscript? Think about the most important results; the important of this information to develop the breeding strategies and improvement of goat reproductive traits!!!
Author Response
A cover letter with responses to the reviewers' comments on animals-518329R1 Dear editor and anonymous reviewers, We greatly appreciate the anonymous reviewers for your careful review and constructive comments (Manuscript ID: animals-518329). We have read comments carefully and tried our best to revise the manuscript (animals-518329R2), and we hope that the revision meet your approval. According to the reviewer’s comments, we mainly made the following modifications. Firstly, we reworte the abstract and highlighted the main research objectives, results and conclusion. Secondly, we modified the sub heading to four parts, for Non-synonymous SNPs, synonymous SNPs, regulatory region SNPs and others mutation within goat GDF9 gene, respectively, and in every part we added the location of SNPs in the new goat reference genome to make it easier for readers to get the information of different SNPs. Thirdly, according to the informantion of the Table S1 (supplementary material), we made two new tables in manuscript, one is for non-synonymous SNPs (Table1) and the other is for synonymous (Table2). In these two tables, it could easy to find the basic information of SNPs, like the SNPs’ location, the frequency of the mutant allele and the effects of the mutant allele on the goat litter size trait. Since the SNPs in the regulatory region of the goat GDF9 gene have hardly been reported before, this part of SNPs information is still listed in the Table S1. Furthermore, we have homogenized the SNPs’ writing thought as soon as we can, which gave the information about the SNPs, and pointed out the effects type of mutation sites on litter size and dicussed its relationships with genotypes, breeds and other factors. In the conclusion, we rearranged the logical order, and basic on summarize and analyze the results of this manuscript, we thought that there were 6 S, A240V, Q320P, V397I, L61L, N121N and L141L, that have ture influence on goat reproductive traits, while other have no effects. we have rechecked and rewrote some sentences in this manuscript to make it easy and clear to understand. Hence, we think it is better than the previous version. Here, we have listed the point-by-point responses to your detailed comments and suggestions (with blue). As follows: Responds to the reviewer’s comments: Reviewer #2: [Comment 1] In line 12, page 1; please add the full address for the corresponding authors, not only the E-mail address. [Response 1] Thanks. According to your suggestion, we added the full address for the corresponding authors as following: *Corresponding authors: Ph.D. Xianyong Lan (lanxianyong79@126.com), College of Animal Science and Technology, Northwest A&F University, Shaanxi Key Laboratory of Molecular Biology for Agriculture, Yangling, Shaanxi 712100, P. R. China, and Ph.D. Haijing Zhu (haijingzhu@yulinu.edu.cn), Shaanxi Provincial Engineering and Technology Research Center of Cashmere Goats, Yulin University, Yulin, Shaanxi 719000, P. R. China and Life Science Research Center, Yulin University, Yulin, Shaanxi 719000, P. R. China. [Comment 2] In line 22, page 1; you can write the non-Synonymous SNPs first and then Synonymous SNPs and consider the order like that in the whole manuscript. [Response 2] Thanks. We revised the order of the SNPs in this sentence and also used this order in the whole manscript. The revised sentence as following: “The available data indicated that non-synonymous SNPs A240V, Q320P, V397I, and synonymous SNPs L61L, N121N, L141L are six “true” effective SNPs for improving goat reproductive traits. ” [Comment 3] In line 42 in the keywords, only mentioned SNP and delete polymorphisms. [Response 3] Thank you for your suggestion. According to your suggestion, we deleted “polymorphisms” in the keywords. [Comment 4] In the introduction part line 47, make it to sentences, you considered goat milk with their important nutritional value because it has less milk fat, and the protein digestion. Then you talk about importance of goat milk in dairy products like cheese and yoghurt…etc. [Response 4] Thank you for your advice. Based on you suggestion, we corrected this sentence into “Goat milk has the characteristics of richer casein micelles, less lactose and higher proportion of small milk fat globules, which make it has greater similarity to human milk and easier for human absorption and utilization. Therefore, goat milk dairy products, like cheese, yoghurt or butter have superior nutritional value for people than other mammalian milk and dairy products [1,2]”. [Comment 5] In the introduction part line 58, please add references after quality. [Response 5] Thank you for you suggestion. According to your suggestion, we gave the reference after this sentence. The specific modification results are as follows: “However, the global goat industry currently faces many problems, the most prominent of which is that the goat products supplies cannot satisfy people’s higher demand in terms of quantity and quality [4]. [4] Lu, C.D.; Miller, B.A. Current status, challenges and prospects for dairy goat production in the Americas. Asian-Australas J Anim Sci 2019, 32, 1244-1255.” [Comment 6] In the introduction part line 70, no space between analysis and identified. [Response 6] Thanks. We added the space between “analysis” and “identified”. [Comment 7] In the introduction part line 75, when you mentioned the name of the gene in the first time, please write the full name and the abbreviation between brackets, then you can used the abbreviation after that in your text. [Response 7] Thanks! According to your suggestion, we added the ful name of genes in this sentence, and the revised results are as follows: “In goats and sheep, many high prolific candidate genes have been identified, like bone morphogenetic protein receptor type 1B (BMPR1B) gene,bone morphogenetic protein 15 (BMP15) gene, growth differentiation factor 9 (GDF9) gene, etc.” [Comment 8] In the introduction part line 80, please specify the breed not talk in general. And the same situation in line 82, which goat breed, please be specific. In general, when you mentioned the name of the reference in the text please write the year also after et al. [Response 8] Thank you for your useful advice. Based on your suggestions, we have added the studied breeds in these two parts. The revised sentences are as following: “The BMPR1B gene, also known as the FecB gene, plays a major role in sheep prolificacy, and several research projects have identified that the mutation Q249R within the FecB gene is highly associated with the ovulation rate in many sheep breeds (eg. Scottish Blackface Merino, Booroola Merino, Garole, Javanese, Mérinos d’ Arles and Small Tailed Han ewes) around the world [13-18]. Conversely, it showed low polymorphism in goats (eg. Black Bengal, Beetal, Barbari, Malabari, Osmanabadi, Ganjam, Jining Grey, Wendeng Dairy and Inner Mongolia Cashmere goats) and had no association with reproductive traits [19-22]” [Comment 9] In line 146, you talked about the different physiological effects which is confusing you said. Which type of confusing, make everything in clear way. [Response 9] Thank you for your suggestion. we have rewritten this part as following: “Until now, a total of 45 GDF9 SNPs have been reported in various goat breeds. However, different studies have shown that the same mutation site has different physiological effects. For instance, the association between one SNP locus and reproductive traits varied with different goat breeds. At the same time, the same allele exhibited positive correlation with high prolificacy in some goat breeds but showed negative correlation in other goat breeds.” [Comment 10] In line 155, were used to detect instead of to detected. [Response 10] Thanks. According to your opinion, we replace “deceted” with “detect” in this sentence. [Comment 11] From line 160 to 165, you mentioned three different number after n, I could not follow you at all. Please in general try to talk and concentrate in the most significant effects of these SNPs on the reproductive traits, which type of effect? And the magnitude of the effect!!! Please try to be specific either used mutant allele or reference allele, sometimes you used major allele (both reference and mutant could be major allele or minor allele in the population, that make me sometimes lost to follow you. [Response 11] Thank you for your advice. First, the different number after n represent the sample size of the same goat breed in different sutdies. For example, in sentence “Ahlawat et al. (2015, 2016) and Maitra et al. (2016) used seven Indian goat breeds, including Black Bengal goats (n = 110/158/158, each number represents the sample size of the each cited researches. The following expression is the same meaning)” , 110 and the first 158 represent sample sizes of the two papers published by Ahlawat et al in 2015 and 2016, respectively, and the last 158 was the sample size of the paper by Maitra et al. in 2016. Second, according to your suggestion, we talked and concentrate the SNPs’ significant effects and the type of the effects on the goat litter size trait, the detailed modification results please refer to line 222-231, line 247-251, and line 267-272, line 281-291 etc. in the revised version. Furthermore, to avoid confusion, we have specific the mutant allele and reference allele rather than major in the revised manuscript. For details, please refer to line 223, line 246, and line 251, etc. in the revised version. [Comment 12] In line 230, please let me know how you draw this conclusion? It has no effect or? [Response 12] Thanks. We revised this sentence and made it more clear to understand. The revised sentences were as following : “Therefore, based on the above studies on goat breeds in China, Iran and India, it can be seen that the Q320P site showed an unstable effect on the goat litter size trait, which would change with the changes of goat breeds. Nevertheless, it is of great significance to find the true role of Q320P via further research with larger sample sizes and a wide range of global goat breeds, as well as the mechanism of Q320P in the reproductive traits is worthwhile.” [Comment 13] In line 236, of the mutant allele instead of mutation. [Response 13] Thanks. According to your suggestion, we replace “mutation” with “mutant”. [Comment 14] In line 242, which type of effect this SNP has in the first-born litter size? [Response 14] Thank you for your advice. The A240V (g.3665C>T) had a positive effect on goat litter size of Henan dairy goats, and we added the discussion of the effect of the frequency of this mutation and the genotypes on goat litter size in the revised manuscript. The specific modifications are as follows: “Interestingly, there also only found two genotypes “CC” and “CT” in these three goat breeds and the frequency of the “T” allele was still low, with values of 0.075, 0.036, and 0.024, respectively; nonetheless, the association analysis showed the A240V significantly affected on the first-born litter size of Henan dairy goats (P < 0.05), heterozygous individuals (genotype CT) had 0.60 kids more than those with the wild homozygote individuals (genotype CC) [53]. Form above studies, it could find there was no mutant homozygotes in 6 goat breeds and the mutation frequency was lower, which might due to the elimination of “TT” individuals from goat breeding over time. Furthermore, “CT” genotype showed greater litter size but only in one breed. Therefore, it is necessary to focus on more breed and a large experimental population to further study the relationship between the A240V and goat reproduction. ” [Comment 15] In line 256, again as I told you before either focus on mutant or reference allele in your manuscript? Which one has a positive effect on the reproductive performance of the goat? [Response 15] Thank you for your suggestions. According to your suggestions, we mainly focus on the mutant allele on the goat reproductive performance, and the L61L ( g.2006C>A) mutation showed negative effect on the litter size of Jining Grey goats, Anhui White goats and Yimeng goats. Therefore, for L61L, the wild type has a positive effect on goat reproductive traits. The revised results are as following: “Meanwhile, association analysis indicated that L61L could significantly affect the litter size of the prolific breed Jining Grey goats, and mutant homozygote “AA” had the lowest litter size (P < 0.05) [63]. Moreover, L61L was detected in Anhui White goats (n = 68), Lubei goats (n = 90) and Yimeng goats (n = 80), and it also had an effect on the litter size of Anhui White goats and Yimeng goats, for which dominant genotypes were consistent with the study of Chu et al. (2011) [54,58]. It can be seen that there was a negative correlation between the L61L and the goat litter size trait, which means it could be used as a useful marker to select the high fecundity individuals in the goat breeding process.” [Comment 16] In line 304, the other mutation in GDF9 gene has a real and significant effect on reproductive traits? If yes in which direction? You talked always about the small sample size if there is no effect for the SNP? Did you recommended special sample size to find association analysis even in different goat populations??? What is your recommendation in general and your conclusion idea after you reviewing all these papers regarding the effect of this gene? What is the next step after to take benefit out of your manuscript? Think about the most important results; the important of this information to develop the breeding strategies and improvement of goat reproductive traits!!! [Response 16] Thank you for you valuable advice! The section of “Other mutation in goat GDF9 gene” mainly focus on some unverified or unreported SNPs, and the effect of these mutations on goat reproductive traits is not clear now. About the sample size, we don't agreed that if the mutation doesn't affect reproductive traits in goats due to the sample size, this only a possibility. But there's no denying that a larger sample size will make your results more accurate and reliablethe, and in recent years, the international sample size of goat research can reach 2000. Furthermore, the sample size of previous studies collected in this study was much smaller than 100, which made us raise doubts about the true role of mutations. It is best to study with a specific sample size to find association analysis in different goat populations, while we can only conduct comparative analysis on the sample size reported by others, but not on the original data. And according to your suggestion, we reduced this account in the revised manuscript as soon as we can. In addition, we resummarized our results in the Conclusion and highlighted the most important results, details please refer to line 392-418 in the revised version. Thanks very much for your comments and attention to our paper. With best Regards! Yours Sincerely, M.D. X.Y. Wang, Ph.D. X,Y. Lan and Ph.D. H, J. Zhu (corresponding author) . College of Animal Science and Technology, Northwest A&F University, Yangling, Shaanxi 712100, China
Round 2
Reviewer 1 Report
Not sufficient improvement to warrant publication.
Author Response
A cover letter with responses to the reviewers' comments on animals-518329R1 Dear editor and anonymous reviewers, We greatly appreciate the anonymous reviewers for your careful review and constructive comments (Manuscript ID: animals-518329). We have read comments carefully and tried our best to revise the manuscript (animals-518329R2), and we hope that the revision meet your approval. According to the reviewer’s comments, we mainly made the following modifications. Firstly, we reworte the abstract and highlighted the main research objectives, results and conclusion. Secondly, we modified the sub heading to four parts, for Non-synonymous SNPs, synonymous SNPs, regulatory region SNPs and others mutation within goat GDF9 gene, respectively, and in every part we added the location of SNPs in the new goat reference genome to make it easier for readers to get the information of different SNPs. Thirdly, according to the informantion of the Table S1 (supplementary material), we made two new tables in manuscript, one is for non-synonymous SNPs (Table1) and the other is for synonymous (Table2). In these two tables, it could easy to find the basic information of SNPs, like the SNPs’ location, the frequency of the mutant allele and the effects of the mutant allele on the goat litter size trait. Since the SNPs in the regulatory region of the goat GDF9 gene have hardly been reported before, this part of SNPs information is still listed in the Table S1. Furthermore, we have homogenized the SNPs’ writing thought as soon as we can, which gave the information about the SNPs, and pointed out the effects type of mutation sites on litter size and dicussed its relationships with genotypes, breeds and other factors. In the conclusion, we rearranged the logical order, and basic on summarize and analyze the results of this manuscript, we thought that there were 6 S, A240V, Q320P, V397I, L61L, N121N and L141L, that have ture influence on goat reproductive traits, while other have no effects. we have rechecked and rewrote some sentences in this manuscript to make it easy and clear to understand. Hence, we think it is better than the previous version. Thanks very much for your attention to our paper. With best Regards! Yours Sincerely, M.D. X.Y. Wang, Ph.D. X,Y. Lan and Ph.D. H, J. Zhu (corresponding author) . College of Animal Science and Technology, Northwest A&F University, Yangling, Shaanxi 712100, China